# Visual Pinwheel Centers Act as Geometric Saliency Detectors

**Haixin Zhong**[1,2]
hxzhong@fudan.edu.cn

**Mingyi Huang**[1,3]
myhuang20@fudan.edu.cn

**Wei P. Dai**[1,5]
weidai@fudan.edu.cn

**Haoyu Wang**[3]
haoyuwang18@fudan.edu.cn

**Anna Wang Roe**[4]
annawang@zju.edu.cn

**Yuguo Yu**[1,2,3,5,*]
yuyuguo@fudan.edu.cn

1. Research Institute of Intelligent Complex Systems, Fudan University.
2. State Key Laboratory of Medical Neurobiology and MOE Frontiers Center for Brain Science, Institutes of Brain Science, Fudan University.
3. Institute of Science and Technology for Brain-Inspired Intelligence, Fudan University.
4. MOE Frontier Science Center for Brain Science and Brain-machine Integration, School of Brain Science and Brain Medicine, Key Laboratory of Biomedical Engineering of Ministry of Education, College of Biomedical Engineering and Instrument Science, Zhejiang University.
5. Shanghai Artificial Intelligence Laboratory.
∗ Corresponding author.

## Abstract

During natural evolution, the primary visual cortex (V1) of lower mammals typically forms salt-and-pepper organizations, while higher mammals and primates develop pinwheel structures with distinct topological properties. Despite the general belief that V1 neurons primarily serve as edge detectors, the functional advantages of pinwheel structures over salt-and-peppers are not well recognized. To this end, we propose a two-dimensional self-evolving spiking neural network that integrates Hebbian-like plasticity and empirical morphological data. Through extensive exposure to image data, our network evolves from salt-and-peppers to pinwheel structures, with neurons becoming localized bandpass filters responsive to various orientations. This transformation is accompanied by an increase in visual field overlap. Our findings indicate that neurons in pinwheel centers (PCs) respond more effectively to complex spatial textures in natural images, exhibiting quicker responses than those in salt-and-pepper organizations. PCs act as first-order stage processors with heightened sensitivity and reduced latency to intricate contours, while adjacent iso-orientation domains serve as second-order stage processors that refine edge representations for clearer perception. This study presents the first theoretical evidence that pinwheel structures function as crucial detectors of spatial contour saliency in the visual cortex.

## 1 Introduction

The seminal work of Hubel and Wiesel revealed orientation-selective columns in the visual cortex of higher mammals [1, 2]. In higher mammals' primary visual cortex (V1), neurons cluster into "pinwheel" structures around singularities [3], unlike in some mammals like rodents, which display "salt-and-pepper" organizations [4] or mini-columns [5]. While there are established theories and experiments for studying the formation of topological organization maps in the visual cortex [6, 7, 8, 9, 10, 11], the functional significance of pinwheel-like columnar organization remains an unresolved question and is even debated [12, 13].

38th Conference on Neural Information Processing Systems (NeurIPS 2024).

Sophisticated visual analyses, such as image pattern extraction [14], pattern symmetry [15], material properties [16], and textures [17], are crucial for understanding complex visual inputs. Imaging and electrophysiological studies have shown that iso-orientation domains (IODs) undergo cross-orientation suppression [18], reducing a neuron's response to its preferred orientation when another orientation is also present in the stimulus [13, 19, 20]. This indicates IODs encoding the linear oriented stimuli, which is crucial for detecting edges and contours [21, 22]. Cross-orientation suppression is believed to facilitate the detection of local discontinuities, such as orientation discontinuities [23, 24, 25], leading to perceptual "pop-out" effects and the perception of illusory contours [24, 26, 27]. In contrast, neurons at pinwheel centers (PCs) exhibit greater selectivity for cross-orientation stimuli [12, 13]. This indicates that PCs respond more effectively to multi-orientation patterns, such as pattern symmetry than IODs [12]. This indicates PCs may contribute to encode more complex contour features. However, PCs are less selective but have longer response latency than IODs for stimulus orientation in the hierarchy process within OPMs when it comes to a single stimulus orientation [13, 19, 28]. Some studies indicate that colors [29], textures [30], darks and lights [31], luminance [32], and mirror symmetry [15] play a role in salient to visual processing. Despite these insights, the functional implications of how neurons within IODs and PCs of pinwheels process complex contour stimuli—potentially affecting stimulus salience for both IODs and PCs—from bottom-up visual inputs remain poorly understood, particularly from a temporal-spatial neural dynamics standpoint.

In response to these challenges, our research contributes the following:

- We propose a novel 2D self-evolving spiking neural network (SESNN) model that investigates the spiking mechanisms behind orientation preference maps (OPMs), spanning from salt-and-pepper organizations in mice to pinwheel structures in cats and macaques. The SESNN uniquely produces sparse codes through local synaptic plasticity during natural image learning, establishing a new benchmark for neural coding strategies.

- PCs act as first-stage processors, detecting natural images and initiating spiking waves to neighboring IODs, which then process as second-stage neurons. This indicates that early processing involves complex contours, not just edge detection.

- PCs react faster to a variety of orientation features than IODs, indicating their function in detecting complex orientations and serving as geometric saliency detectors. This suggests PCs have an evolutionary advantage due to self-organized pinwheel structures, which improves their ability to process complex contours.

## 2 Results

### 2.1 Visual overlap underlying pinwheels emergence

Our SESNN model generates diverse OPMs, from salt-and-peppers to pinwheel structures, by adjusting the visual overlap metric $\varepsilon$. This metric, crucial for the variety of visual topologies across species, is shown in Fig. 1a to produce pinwheel structures at high overlap, akin to those in cats and macaques, while low overlap results in salt-and-pepper organizations, typical of mice or rats. High overlap also enables cortical neurons to sample natural scenes more frequently, aiding in generating high-resolution images during decoding [7, 33].

Fig. 1a shows how visual input overlap levels from 9 to 15 pixels affect V1 orientation selectivity maps in the model. The top panel illustrates a higher overlap (15 pixels), and the middle, a lower overlap (12 pixels). This comparison reveals the impact of stimulus overlap on pinwheel density and layout in the visual cortex. Below the threshold (10 pixels in our case), salt-and-pepper patterns form, as the bottom panel indicates. Thus, 9 pixels of overlap are excluded from pinwheel analysis, as shown in Fig. 1b-d.

We quantitatively analyze the OPMs shown in Fig. 1a with several metrics [7, 34]:

**Pinwheel counts**, defined as the number of PCs, can be measured by 2D fast Fourier transform [35], which are located at the intersection of the real and imaginary components that equal 0 [34]. It exhibits a decreasing trend as the visual input overlap increases (illustrated in Fig. 1b), suggesting that a greater overlap in the visual field may lead to a reduction in the number of discrete pinwheel structures.

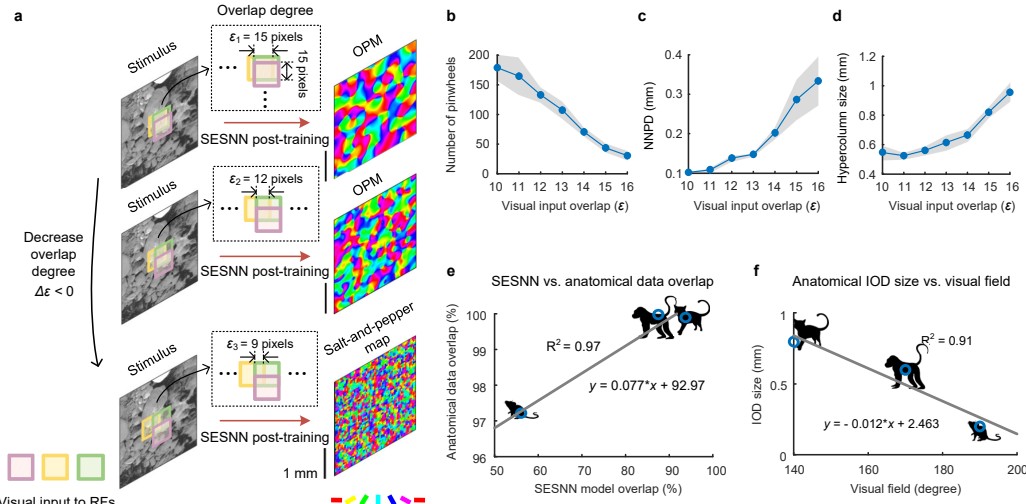

Figure 1: Receptive field (RF) visual overlaps underlying the emergence of OPM and the salt-and-peppers are revealed via our SESNN model. **a**. Modifying the overlap parameter ($\varepsilon$) among neighboring neurons receiving ($16 \times 16$ pixels) visual inputs from natural images influences the dimensions (e.g., **b**. Pinwheel counts, **c**. Nearest-neighbor pinwheel distance, **d**. Hypercolumn size) of pinwheel structures and salt-and-pepper organizations. (Lines: mean. Shaded area: SD.) **e**. Comparing the SESNN model overlap percentage ($\frac{\varepsilon}{s_{\mathrm{RF}}} \times 100\%$) with actual anatomical data overlap percentages ($\varepsilon'_{\mathrm{percentage}}$) in various species (mice, cats, and macaques). **f**. Relationship between the IOD size and the extent of the visual field in anatomical data (mice, cats, and macaques).

**The nearest-neighbor pinwheel distance (NNPD)** in millimeter (mm) unit is defined as the distance between the two nearest PCs. The increasing trend of visual input overlap expands the distance between neighboring pinwheels (Fig. 1c).

**The size of hypercolumns** (mm) is defined with periodicity measured by 2D fast Fourier transform and also increases with the visual input overlap (shown in Fig. 1d). This paper does not account for left- and right-eye dominance columns, so the hypercolumn size is defined as the full 180° cycle of repeating column spacing ($\Lambda$) (mm).

It's noteworthy that pinwheel density is not included as a metric in our analysis. This omission is because the observed pinwheel density, irrespective of the hypercolumn size, approaches $\pi$ pinwheels/$\Lambda^2$, conforming to topological constraints [34, 35].

Our findings emphasize the importance of overlap degrees (Fig. 1a). Greater overlap (e.g., $\varepsilon_1 = 15$ pixels) fosters stronger local clustering, leading to larger hypercolumn sizes, fewer pinwheels, and longer NNPDs, versus lower overlap (e.g., $\varepsilon_2 = 12$ pixels). Minimal overlap (e.g., $\varepsilon_3 = 9$ pixels), yields weak clustering, resembling salt-and-pepper organizations. This suggests that shared input among V1 neurons significantly influences OPM and salt-and-pepper formation. We obtain the anatomical data overlap using Eq. 3 and observe a strong positive correlation ($R^2 = 0.97$) between the SESNN model and species' visual RF overlaps (mouse, cat, macaque) (Fig. 1e). This relationship highlights the overlap index's key role in spatial organization within orientation maps. The model's predictions on IOD sizes and visual field extent (Fig. 1f) align with empirical data [7], confirming the SESNN model's robustness in simulating neuroanatomical organization and the biological development of orientation maps.

## 2.2 Spatial-temporal distributed spiking waves propagate within pinwheels

V1 neurons stimulated by natural images primarily fire within pinwheel structures, particularly within and around PCs (Fig. 2a-b). This pattern is especially pronounced in higher mammals with large IODs, such as macaques and cats.

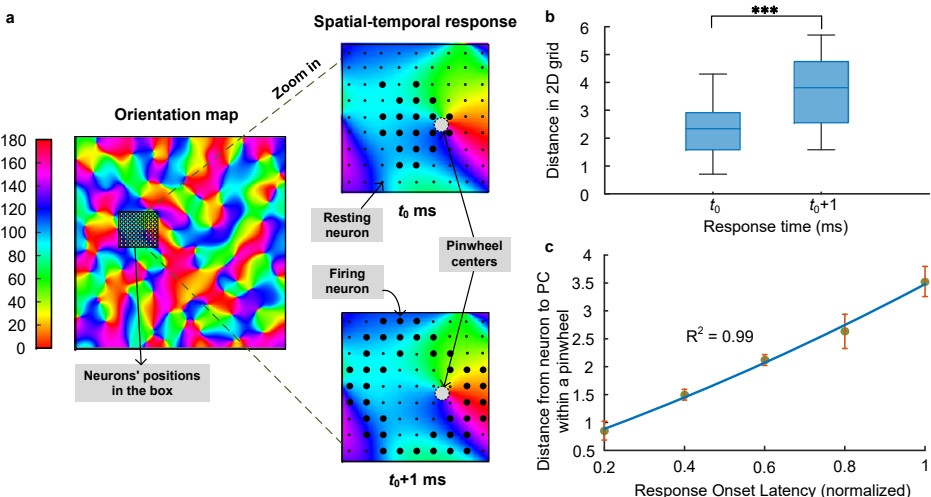

Figure 2: Spatial-temporal response pattern within pinwheels. **a**. This figure displays the neuronal responses on OPM with a large IOD in a pinwheel structure. The neurons that fire at time $t_0$ are shown as large black dots at PC, and they expand towards the periphery at time $t_0+1$, also denoted as large black dots. The other small dots represent resting neurons. **b**. Distance between firing neurons and the PC at time $t_0$ and $t_0+1$. **c**. This panel shows the response onset latency of neurons and the mean distance ($\pm$ SD) between these neurons within a pinwheel. The distance is measured as the Euclidean distance within a 2D grid, simulating the structure of a 2D V1 area. (Significance: ***p<0.001, Mann-Whitney U test.)

We define the response onset latency as 1 ms for the initial discharge from pinwheel structures, with subsequent firings occurring at 2 ms, based on a 1 ms time unit. Stimulated by natural images, the discharges start at the PCs and exhibit pronounced diffusion within the IODs sequentially, depending on their distance from the center, as suggested in Fig. 2c.

## 2.3   Visual bottom-up saliency detection: functional role of pinwheel in geometric encoding

In this section, we investigate whether pinwheel structures respond distinctly to salient features in input images. The ground truth boundary from the BSDS 500 dataset [36] used as binary input represents geometric complexity (edges and curves) (Fig. 3a). The complexity is measured by calculating the local pixel entropy using sliding windows, with a 15×15 pixel neighborhood to assess pixel value dispersion in the binary images. The computation adheres to the following equation:

$$H(i,j) = -\sum_{k=0}^{L-1} p(m_k) \log_2 p(m_k), \tag{1}$$

where $H(i,j)$ denotes the entropy at pixel position $(i,j)$ in the entropy map, $L$ the count of distinct gray levels within the local neighborhood around pixel $(i,j)$, and $m_k$ the $k$th gray level within this specified neighborhood. A large entropy value reflects great unpredictability or complexity in the pixel values, signifying a highly variable pixel value distribution. Conversely, a low entropy value indicates a high degree of predictability, less variation, and reduced complexity in the contours of pixel values. In addition, the saliency map of images is generated based on the classical methodology [37].

Furthermore, we propose a bimodal ratio analysis to compute the orientation bimodal ratio (OBR) to indicate a neuron's orientation tuning curve as either unimodal (single peak) or perfectly bimodal (two peaks of equal strength). This analysis focuses on identifying the peaks in the orientation tuning curve and quantifying their relative strengths.

$$OBR = \frac{2 \cdot \min(R_1, R_2)}{R_1 + R_2}, \tag{2}$$

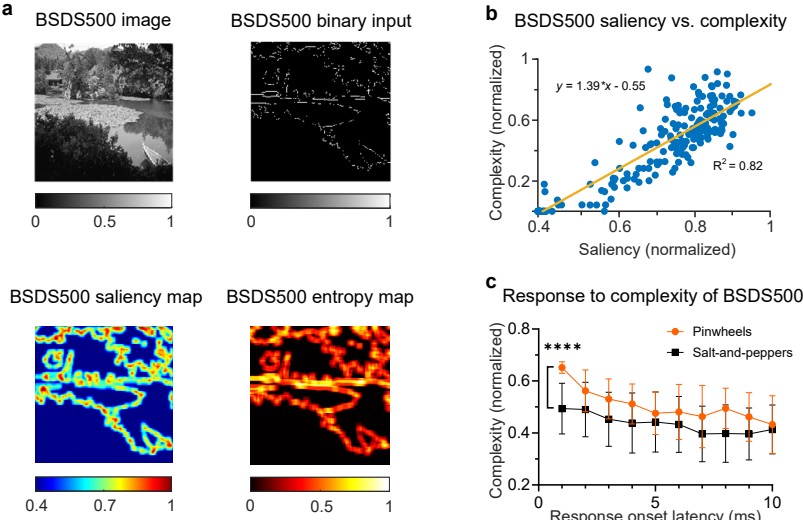

Figure 3: Pinwheel structures in V1 exhibit geometric properties. **a**. A BSDS 500 grayscale image displays boundaries, saliency, and entropy maps. **b**. Natural images show a positive correlation between saliency and entropy. **c**. Neuronal response onset latency from pinwheels and salt-and-peppers relates to structural complexity, measured by local pixel entropy. (Data: mean ± SD, significance: ****p<0.0001, Welch's t-test.)

where $R_1$ and $R_2$ represent the normalized firing rates corresponding to the strengths of the two most pronounced peaks in the orientation tuning curve. The OBR ranges from 0, denoting unimodality, to 1, indicating perfect bimodality in the neuron's orientation tuning.

A positive correlation is observed between the saliency map and the geometrical complexity of the BSDS500 dataset (Fig. 3b), demonstrating that higher geometrical complexity correlates with increased saliency. Significantly, in response to the stimulus shown in the BSDS500 image (Fig. 3a), pinwheel structures primarily activate in areas of high contour complexity (regions with the highest saliency in this binary image), which is a response pattern have not been observed in salt-and-peppers (Fig. 3c).

To confirm the disparity in contour complexity responses between pinwheel and salt-and-pepper organizations, we design a star-like binary input (depicted in Fig. 4a), including four identical entities to negate the impact of neuronal positioning within the SESNN model. This approach reaffirms the saliency-complexity correlation (Fig. 4b) and the priority of pinwheel activation over salt-and-peppers in response to heightened complexity (Fig. 4c).

Findings show that PCs exhibit enhanced saliency detection and significantly faster response times than IODs, indicating that PCs respond more quickly and sensitively to geometrically complex stimuli, while IODs are slower and react to simpler geometrical stimuli (see Fig. 4d). Both saliency and latency measurements are normalized to a 0-1 scale for comparison.

The enhanced saliency detection of PCs is due to the complex orientation preference in RFs. As addressed in Fig. 4e, the ordinate represents the OBR, reflecting that neurons near PC generally exhibit bimodal orientation tuning curves with near-equal peak strengths while there is a primary peak and a secondary peak at a relatively far position from PC ($x = 2$). And the secondary peak is nearly absent at the IODs level ($x = 3$), leading to an OBR close to 0. Salt-and-peppers, however, show less variation, maintaining a consistent OBR.

In conclusion, PCs demonstrate selectivity for more intricate orientations. This is experimentally supported by [13, 38], who suggest that PCs are particularly sensitive to specific geometric configurations, such as T junctions. Characterized by multiple orientations and an OBR nearing 1 (Fig. 4e), these neurons tend to initiate action potentials in response to complex orientations. Consequently,

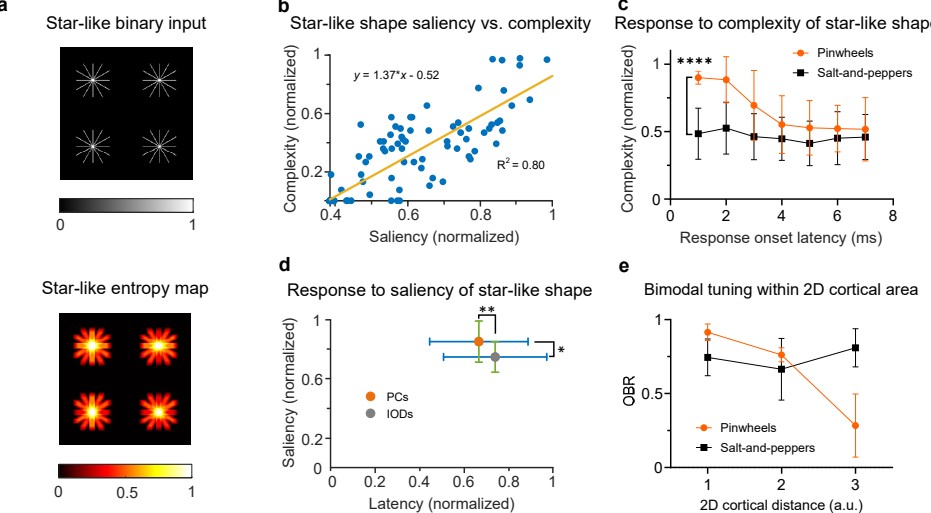

Figure 4: Geometric properties emergence in PCs of V1 on star-like patterns. **a**. We introduce artificial star-like patterns to assess neural response to complexity. **b**. Star-like images show a link between saliency and entropy. **c**. Neuron response times in PCs and salt-and-peppers reduce with lower entropy. **d**. The analysis compares PCs and IODs for saliency and response to star-like patterns; the inset details saliency and latency. **e**. OBR varies across cortical distance; the red line marks pinwheels, and the black line, salt-and-peppers (an arbitrary point for salt-and-peppers). (Data: mean ± SD, significance: **p<0.01, ***p<0.001, ****p<0.0001, Welch's t-test.)

this leads to pinwheels being the first to respond. In contrast, neurons in salt-and-peppers do not exhibit a similar responsiveness to complex orientations as observed in pinwheels.

## 3 Methods

### 3.1 The architecture of SESNN model

Our SESNN model is a two-dimensional network of excitatory (E-) and inhibitory (I-) leaky integrate-and-fire neurons (LIF) (5), stimulated by whitened natural images to mimic the LGN's functions of contrast normalization and edge enhancement without complex modeling [43, 44, 45]. We use 160 whitened natural images as the training dataset, normalized to zero mean and uniform variance, derived from 20 base images (512×512 pixels) [45, 46]. To capture orientation details, each of the base images undergoes a 90-degree clockwise rotation and flip, creating 8 variations per original.

The configuration features E- and I- neurons in recurrent networks with periodic boundary conditions (PBCs) (Fig. 5b), simulating a continuous 2D cortical surface. The neural connectivity at initialization is depicted in Fig. 5c (see Eq. 11). Moreover, the connection strengths in the well-trained model align closely with the experimental finding ([47]). Under natural image stimuli, the SESNN forms single neuron RFs and population-level pinwheel structures in the OPM (Fig. 5e-f). To validate the model, we compare its evolution from randomness to organized states against biological data from macaque pinwheel structures and a baseline model [42], using metrics such as pinwheel density (pinwheels/ $\Lambda^2$), NNPD (mm), and hypercolumn size (mm) [7, 34, 35] (Fig. 5f and Table 1).

Table 1: SESNN pinwheels (mean ± SD) vs. macaque pinwheels.

| Metric | E-I baseline | SESNN model | Macaque |
|---|---|---|---|
| Pinwheel density (pinwheels/$\Lambda^2$) | $\sim 2.941$ | $3.175 \pm 0.397$ | $\sim 3.327$ |
| NNPD (mm) | N/A | $0.277 \pm 0.043$ | $\sim 0.242$ |
| Hypercolumn size (mm) | N/A | $0.839 \pm 0.054$ | $\sim 0.760$ |

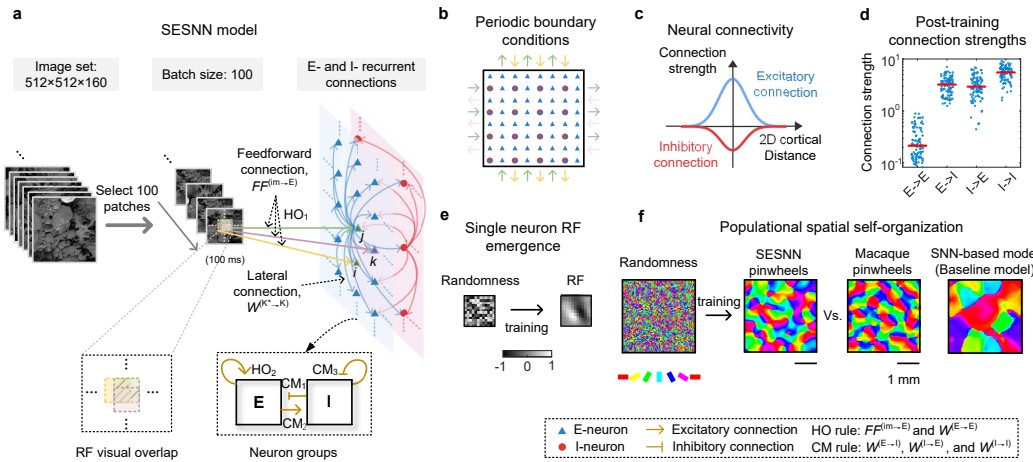

Figure 5: Architecture of proposed SESNN model. **a**. The SESNN model comprises 4,900 E- and 1,225 I- neurons [39, 40, 41]. It processes 160 natural images (100 patches each), presenting each 512×512 pixel patch to E-neurons for 100 ms with input overlap. The Feed-forward and E-E connections adhere to the Hebbian-Oja (HO) rule; others follow the Correlation Measuring (CM) rule. **b**. E- and I-neurons are spatially arranged with periodic boundaries, sharing coordinates with connected boundaries as per diagram arrows. Identical connections are marked by same-color arrows. **c**. Initial weights are Gaussian distributed. **d**. Post-training connection strengths are depicted, with medians in red. **e**. RF emerges after training. **f**. Post-training spatial organization is compared among the SESNN model's OPM, macaque V1, and an SNN-based model [42], with color bars for orientation and a 1 mm scale bar on the cortical surface.

### 3.2 Experiment-data-justified overlapping visual fields among nearby neurons

In each trial, E-neuron processes 100 different 16×16 patches for 100 milliseconds each, randomly selected from the training dataset to serve as RFs (see Fig. 5a middle and right panels). It is assumed that these visual inputs overlap on the retina (Fig. 5a, middle panel and its inset). To reflect biological conditions, we perform a statistical analysis based on data from cats, macaques, and mice (Table 2), calculating average overlaps of 99.90% for cats, 99.98% for macaques, and 97.23% for mice using (Eq. 3). These results closely align with our SESNN model's configurations (refer to Fig. 1e). RF size in V1 is more related to resolution than orientation map formation. In macaque V1, RF size increases more than tenfold from fovea to periphery, while orientation map properties show little variation [48, 49]. Our study does not focus on RF size variations across the retina, as we expect minimal effects from these shifts across species, provided the overlap remains constant. Since the fovea is key for detailed visual information, we use V1 RFs in the area centralis to modeling.

We propose the visual input overlap metric $\varepsilon'_{\text{percentage}}$, which is defined as follows:

$$\varepsilon'_{\text{percentage}} = \frac{\sqrt{\rho_{\text{V1}} S_{\text{unit}}} - \frac{L_{\text{unit}} M^{-1}}{s'_{\text{RF}}}}{\sqrt{\rho_{\text{V1}} S_{\text{unit}}} - 1} \times 100\% \tag{3}$$

where $S_{\text{unit}}$ represents the unit cortical area mm$^2$, $s'_{\text{RF}}$ denotes the size of the RF in V1 with unit in degrees, $\rho_{\text{V1}}$ represents the density of neurons in V1, $L_{\text{unit}}$ represents the unit of cortical spacing length in unit of mm, and $M$ refers to the cortical magnification factor (CMF) (mm/degree). We consider only an effective cortical layer composed of output neurons. This is because the apparent overlap within a vertical cortical column primarily contributes to intermediary processing stages for the same input. Therefore, such overlaps should not be conflated with overlaps in the input space.

### 3.3 Neural dynamics

E-neurons receive stimuli from natural images as well as noise $\mathcal{N}(0, 0.04)$ from other brain areas (noise term). I-neurons indirectly receive natural image stimuli by adjusting E-neurons. The neural spiking dynamics are modeled using biologically inspired LIF neurons, incorporating refractory

Table 2: Comparative anatomical data of the retina and V1 across three species. **a**. This table includes three diverse species, encompassing both primates (e.g., macaques) and non-primates (e.g., mice and cats). **b**. V1 neuron density (neurons/mm$^2$) within 2D surface. **c**. Size of V1 RF in area centralis (deg). **d**. Peak CMF (mm/deg) of V1 in area centralis.

| (A)SPECIES | (B)V1 NEURON DENSITY | (C)V1 RF SIZE | (D)PEAK CMF |
|---|---|---|---|
| CAT | $\sim 99,200$ [33] | $\sim 1.0$ [50] | $\sim 1.90$ [51] |
| MACAQUE | $\sim 243,000$ [33] | $\sim 0.2$ [52] | $\sim 18.18$ [52] |
| MOUSE | $\sim 86,600$ [33] | $\sim 4.0$ [53] | $\sim 0.03$ [54] |

periods and adaptive firing thresholds [55]. The neural dynamics are iteratively formulated as follows:

$$u_i^{(\mathrm{K})}(t+1) = u_i^{(\mathrm{K})}(t)e^{-\frac{\eta}{\tau^{(\mathrm{K})}}} + h_{\mathrm{K}}(i)\sum_j \mathrm{FF}_{ij}^{(\mathrm{image}\rightarrow\mathrm{E})}X_j \tag{4}$$

$$+ \sum_{\mathrm{K}^*}\sum_j \beta_{ij}^{(\mathrm{K}^*\rightarrow\mathrm{K})} \cdot W_{ij}^{(\mathrm{K}^*\rightarrow\mathrm{K})} \cdot z_j^{(\mathrm{K}^*)}(t) + \mathrm{noise},$$

$$h_{\mathrm{K}}(i) = \begin{cases} 1, & \text{if } i \text{ is an E-neuron ID,} \\ 0, & \text{if } i \text{ is an I-neuron ID,} \end{cases} \tag{5}$$

$$\Delta\theta_i^{(\mathrm{K})} \propto p_i(z_i^{(\mathrm{K}^*)}=1) - p_i^{(\mathrm{K})}, \tag{6}$$

where $i = 1, 2, \ldots, N$th (the neuron IDs of E-neurons and I-neurons).

In neural dynamics equation, $u_i^{(\mathrm{K})}(t)$ denotes the membrane potential of neuron $i$ at time $t$, applicable to neurons of class K, which includes E- and I- neuron groups. The membrane time constant, symbolized by $\tau$ in the resistor-capacitor circuit, governs the decay rate of the membrane potential in individual neurons. Notably, inhibitory neurons are configured to fire more rapidly than excitatory neurons [44, 56]. This setup reduces reconstruction error and hastens system convergence, leading to a more efficient and accurate representation of input stimuli. See the section A.2 for detailed neural dynamics.

### 3.4 Hebbian Learning in SESNN

The learning rules consist of the HO rule [43] for input weight adjustments and the CM rule [44, 45] for intra-network weight changes (Fig. 5a-c). These facilitate adaptive synaptic weight adjustment based on firing pattern correlations, emulating a key learning mechanism in biological neural networks.

The formula for these adjustments is given by:

$$\mathrm{HO}: \Delta W_{ij}^{(\mathrm{K}^*\rightarrow\mathrm{K})} \propto y_i x_j - y_i^2 W_{ij}^{(\mathrm{K}^*\rightarrow\mathrm{K})}, \tag{7}$$

$$\mathrm{CM}: \Delta W_{ij}^{(\mathrm{K}^*\rightarrow\mathrm{K})} \propto y_i x_j - \langle y_i \rangle \langle x_j \rangle \left(1 + W_{ij}^{(\mathrm{K}^*\rightarrow\mathrm{K})}\right), \tag{8}$$

where $x$, $y$ denote the spike rates of presynaptic and postsynaptic neurons, respectively, with $\langle \cdot \rangle$ denotes the lifetime average. After each stimulus presentation of 100 ms, we calculate the network's neuronal instantaneous spike rates using exponential moving averages (EMAs), which aggregate spikes over time to reflect recent activity (see section A.1). Lifetime averages, also computed as EMAs, are crucial for homeostatic stability, helping to modulate neuronal properties or synaptic strengths for consistent activity. See the section A.2 for the hyperparameters.

Our SESNN model reflects experimental findings [57, 47] by representing V1 pyramidal neurons with weaker synaptic strengths, essential for preventing over-excitation and maintaining neural balance. We apply the HO rule [43] to E-E connections with a normalization factor to keep synaptic weights between 0 and 1, while stronger lateral E-I connections under the CM rule lack this normalization [45, 44]. Post-training synaptic strengths are depicted in Fig. 5d. Stabilizing neural network training requires careful learning rate adjustment. A slower rate for E-E connections compared to others is crucial to prevent E-neuron over-excitation, aligning with empirical data [57, 47, 58].

The HO and CM rules facilitate LTP and LTD mechanisms, common in rate learning rules that do not require precise spike timing. We selected these rules for their ease of tuning and ability to stabilize recurrent excitation.

# 4 Related works

**Functional roles of pinwheel structure can be revealed by SESNN model** The classical self-organizing map model [8] and other computational approaches like on-off models [6, 7, 9, 59] and related ANNs [10, 60, 61] lack the dynamic and temporal fidelity needed to realistically simulate the emergence of pinwheel structures in the visual cortex. To address these shortcomings, we propose the novel SESNN model, integrating retinotopy data [33, 50, 52, 53], detailed morphological data [62, 63, 64], and CMF [51, 52, 54] to enhance biological fidelity. The SESNN model effectively simulates macaque cortical organization and pinwheel development within OPMs (Fig. 5f). Furthermore, our investigations reveal that the degree of overlap—reflecting similar feed-forward inputs from identical RGCs to neighboring neurons—positively correlates with the retino-cortical mapping ratio [6], aiding in distinguishing between different V1 organizational patterns.

**PCs and IODs in neural processing hierarchies** Our findings show that PCs and IODs exhibit distinct neural activity waves, leading to varied responses to contour complexity from spatial-temporal dynamics: PCs react first to complex contours, having more multi-orientation selective neurons (Fig. 4e, $x = 1$) before activity spreads to IODs, which process simpler edges (Fig. 4d and e, $x = 3$). PCs display a stronger correlation with contour saliency, indicating a heightened role in processing visual stimuli over IODs. In rodents with salt-and-pepper organizations, contour saliency is less pronounced (Figs. 3c and 4c). While PCs are thought to indicate higher-order processing due to delayed response [13, 28], this is likely due to the nature of the stimuli. Studies reveal IODs show cross-orientation suppression under complex stimuli [12], unlike PCs with broader tuning. The SESNN model illustrates a preference for complex stimuli in PCs and simple stimuli in IODs, with activity propagating from PCs to IODs upon encountering complex contours (Fig. 4d and e).

**PCs as geometric saliency detectors** The SESNN model reveals PCs have broader orientation tuning and less selectivity for complex contours, unlike IODs, which show sharper tuning and cross-orientation suppression, preferring simpler edges ($x = 3$ in Fig. 4e) [12, 13, 19, 58, 65, 66, 67]. PCs' excitation leads to reduced cross-orientation suppression. With binary input, PCs correlate more positively with contour complexity than IODs (Figs. 3b and 4b), making them more salient in processing visual stimuli. This differs from rodents with salt-and-pepper organizations that lack distinct contour complexity saliency (Figs. 3c and 4c). Prior studies [12, 13, 28] suggest PCs have delayed response latency, indicative of higher-order processing. This arises from using drifting grating stimuli that activate IODs more readily. Koch et al. [12] note that IODs show cross-orientation suppression under complex stimuli, narrowing their tuning, unlike PCs. However, these studies omit temporal neural data within pinwheel structures. The SESNN model supports physiological findings that IODs and PCs favor single and complex orientation stimuli, respectively.

# 5 Conclusion and limitations

The advantages of pinwheel structures in visual representation and encoding are not fully understood. To address this, we develop a two-dimensional SESNN model that incorporates Hebbian-like plasticity and empirical morphological data. This model evolves to function as localized, bandpass filters, enhancing its responsiveness to a range of orientations and complex spatial textures in natural images. Our findings reveal that neurons within pinwheel structures respond more effectively to these textures, with stronger and quicker reactions than those in salt-and-pepper configurations. Specifically, PCs act as first-order stage processors with heightened sensitivity and reduced response latency to intricate contours, while IODs function as second-stage processors, refining edge representation for greater clarity. This advanced processing capability of pinwheel structures, particularly in detecting spatial contour saliency, not only deepens our understanding of visual processing in higher mammals but may also inform new strategies for visual saliency algorithms in computational models.

Using sliding windows, local entropy assesses variation and complexity in spatial distributions by capturing local intensity changes, indirectly reflecting geometric complexity through edges, corners, and patterns. Since this method cannot directly measure geometric shapes, we verify the use of the Ramer-Douglas-Peucker algorithm to approximate and directly measure geometric structures (refer to section A.5) [68]. This algorithm simplifies shape contours by reducing vertices while preserving the overall form. The resulting polygon will allow us to calculate the distribution of edge lengths and angles, with geometric entropy defined as the sum of these entropy values. In future studies, we will utilize the Ramer-Douglas-Peucker algorithm to enhance our geometric analysis by identifying

and measuring the complexity of specific structural features, such as junctions, sharp corners, and textures, which are essential in complex visual scenes.

## Acknowledgments

We gratefully acknowledge the support from the Science and Technology Innovation 2030 - Brain Science and Brain-Inspired Intelligence Project (2021ZD0201301), the National Natural Science Foundation of China (U20A20221, 12201125, 12072113), the Shanghai Municipal Science and Technology Committee of Shanghai outstanding academic leaders plan (21XD1400400), the Yang Fan plan (22YF1403300), and the China Postdoctoral Science Foundation (2023M740724).

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

# A Appendices

## A.1 Exponential moving average

We compute the network's neuronal instantaneous spike rates as exponential moving averages (EMAs), which accumulate spikes over time (see Eq. 9). EMAs are utilized to track recent neuronal activity levels. Concurrently, lifetime average values are also calculated using EMAs, which are crucial for maintaining homeostatic stability. This method helps stabilize the neural network by adjusting neuronal properties or synaptic strengths to sustain consistent activity levels over time.

$$x_j(t) = (1 - \zeta)x_j(t-1) + \zeta \cdot z_j(t), \tag{9}$$

where $\zeta = 1 - e^{-\frac{1}{10}}$, indicating that the 10 ms is a temporal window of the moving average weighted with exponential decay. The initialization of $x_j$ is 0. The exponential moving average is calculated dynamically and updated along with synaptic weights.

$$\langle x_j \rangle := (1 - \xi) \cdot \langle x_j \rangle + \xi \cdot \overline{x}_j, \tag{10}$$

where $\xi = 1 - e^{-1}$. It is dynamically updated to ensure the sum of the weights remains constant over time.

## A.2 Detailed parameters and connectivity settings for the model

**Detailed neural dynamics:** The feed-forward connection, labeled $\text{FF}_{ij}^{(\text{image}\to\text{E})}$, links pixel $X_j$ of the whitened image patch to excitatory (E-) neuron $i$. $W_{ij}^{(\text{K}^*\to\text{K})}$ signifies the synaptic weight from neuron $j$ of neuron class K$^*$ to neuron $i$ of neuron class K, with its sign determined by the connection type, described as $\beta_{ij}^{(\text{K}^*\to\text{K})}$(the neuron receives E-connections, set as +1; conversely, the neuron receives inhibitory (I-) connections, the sign is set as -1). $z_j^{(\text{K}^*)}(t)$ indicates the spike output of neuron $j$ at time $t$. Upon reaching the spike threshold $\theta$ (initialized as 2), a spike is emitted, $z_j^{(\text{K}^*)}(t)$ is set to 1, then the membrane potential is reset to 0 mV, remaining so until the refractory period (3 ms) concludes. Within primary visual cortex (V1), homeostatic plasticity [34, 55] ensures neural activity stability by dynamically adjusting the firing threshold $\theta$. This adjustment is based on the deviation of the current firing rate $p_i(t)$ from the target rates $p_i^{(\text{K})}$ ($p^{(\text{E})} = 2$, $p^{(\text{I})} = 4$), as outlined in Eq. 6 [55]. We assign $\tau^{(\text{E})} = 10$ ms for E-neurons and $\tau^{(\text{I})} = 5$ ms for I-neurons. To enhance computational efficiency, we set the time step to 1 ms.

**Hyperparameters:** For the synaptic plasticity, learning rates are $\eta_{\text{FF}} = 0.2$ (image to E-neurons), $\eta_{\text{EE}} = 0.01$ (E- to E-neurons), $\eta_{\text{EI}} = 0.7$ (I- to E-neurons), $\eta_{\text{II}} = 1.5$ (I- to I-neurons), and $\eta_{\text{IE}} = 0.7$ (E- to I-neurons), while the neural connectivity parameters are $\alpha_{\text{max,E}} = 1.0$ (E- max weight), $\alpha_{\text{max,I}} = 0.5$ (I- max weight), $\sigma_{\text{EE}} = 3.5$ (E-E coupling range), $\sigma_{\text{EI}} = 2.9$ (E-I coupling range), $\sigma_{\text{IE}} = 2.6$ (I-E coupling range), and $\sigma_{\text{II}} = 2.1$ (I-I coupling range).

**Neural connectivity within 2D cortical area:** E- and I- neurons are arranged symmetrically on a two-dimensional lattice, as illustrated in Fig. 5b. Periodic boundary conditions are employed to mimic the large number of neurons in the actual V1 cortical surface. Specifically, neurons at the boundary are connected to neurons at corresponding symmetric positions on the opposite boundary. The initial connection weights between neurons are modeled by a Gaussian function of their distance (see Fig. 5c), which can be expressed as:

$$W_0^{\text{K}^*\to\text{K}}(i, j) = \alpha_{\text{K}^*} \times \exp\left(\frac{-d(i, j)^2}{2\sigma_{\text{K}^*}^2}\right). \tag{11}$$

In this equation, $d(i, j)$ represents the Euclidean distance from neuron $i$ to neuron $j$ in a grid, $\alpha$ determines the maximum connection weight, which is set to $\alpha_{\text{EE}} = 1$, $\alpha_{\text{EI}} = 1$, $\alpha_{\text{IE}} = 0.5$, $\alpha_{\text{II}} = 0.5$, and $\sigma$ governs the rate at which the weight decays with distance. The synaptic types predominantly determine the parameters for this connection weight distribution function. To accurately replicate the neuronal architecture of V1 in macaques. The connectivity radiuses, denoted by $\sigma$, are set to $\sigma_{\text{EE}} = 3.5$, $\sigma_{\text{EI}} = 2.9$, $\sigma_{\text{IE}} = 2.6$, $\sigma_{\text{II}} = 2.1$. These values are based on anatomical data indicating that the axon length scales of E- and I-neurons are approximately 200 $\mu$m and 100 $\mu$m, respectively, while the dendrite length scales are around 150 $\mu$m for E-neurons and 75 $\mu$m for I-neurons in the V1 [62, 63, 64]. We prune any connection strengths below a threshold of 0.01 to maintain computational efficiency and biological plausibility.

### A.3 Anatomical data integration

**Neural connection data**

The experimental subjects include six adult cats with unknown genders, with data sourced from research by Armen Stepanyants et al.[63]; and eight macaques, aged 5-11 years, including six males and two females, with data sourced from research by Joseph Amatrudo et al.[64].

**Neuronal synaptic plasticity**

The subjects are rats aged 14-16 days, with unknown gender and quantity, with data sourced from research by Holmgren et al.[57]; transgenic mice, with unknown quantity and gender, with data sourced from research by Hofer et al. [47].

**Retinal-V1 topological projection data**

Receptive field (RF) data: V1 neuron counts for macaques, cats, tree shrews, ferrets, mice, rats, and gray squirrels respectively come from Tehovnik et al. [69] (subjects: 3 macaques, unknown gender and age), Scholl et al. [50] (subjects: cats, unknown gender and age), Veit et al.[70] (subjects: 9 male and 7 female tree shrews, aged 3-8 years), Huberman et al.[71] (subjects: 8 ferrets, unknown gender and age), Niell et al.[53] (subjects: mice, aged 2–6 months, unknown gender), Foik et al.[72](subjects: 21 rats, unknown gender and age), and Hall et al.[73] (subjects: 17 gray squirrels, unknown gender and age). V1 neuron density: Neuron density data for macaques, cats, mice, rats, and gray squirrels come from Srinivasana et al.[52] (subjects: unknown gender and age); tree shrew, ferret, and gray squirrel density data respectively come from Weigand et al.[74].

**Cortical magnification factor**

Cortical magnification factor (CMF) data for macaques, cats, tree shrews, ferrets, mice, rats, and gray squirrels are sourced from Tehovnik et al.[69] (subjects: 3 macaques, unknown gender and age), Veit et al.[70](subjects: cats, unknown gender and age), Bosking et al.[48] (subjects: tree shrews, unknown gender and age), Rockland et al. [75] (subjects: 9 ferrets, female, unknown age), Beest et al.[54] (subjects: 28 mice, 11 males and 17 females, ages 2-14 months), Keller et al.[76] (subjects: male rats, age 3 months), and Hall et al.[73] (subjects: 17 gray squirrels, unknown gender and age).

Additionally, the anatomical data concerning inter-ocular distances are obtained from Najafian et al. [7].

### A.4 Unveiling species-specific factors distinguishing pinwheels and salt-and-peppers

#### A.4.1 Anatomical data suggests RFs density underlying V1 organizations

Table 3: Comparative anatomical data of the retina and V1 across species.

| **a.** Species (mean) | **b.** Retina ($mm^2$) | **c.** V1 size ($mm^2$) | **d.** V1 neurons density (neurons/$mm^2$) | **e.** V1 RF size in area centralis (deg) | **f.** RFD $((c) \times (d)/(b))$ (RFs/$mm^2$) |
|---|---|---|---|---|---|
| Macaque | 636[6] | 1,090[33] | 243,000[33] | 0.2[52] | 416,462.26 |
| Cat | 510[6] | 380[6, 33] | 99,200[33] | 1.0[50] | 73,913.73 |
| Tree shrew | 122[6, 77] | 73[6, 33] | 192,800[74] | 2.0[70] | 115,363.93 |
| Ferret | 83[6, 75] | 78[33] | 95,813[74] | 3.0[71] | 90,041.13 |
| Mouse | 15[6] | 2.5[33] | 86,600[33] | 4.0[53] | 14,433.33 |
| Rat | 52[6, 78] | 7.1[33] | 90,800[33] | 3.0[72] | 12,397.69 |
| Gray squirrel | 205[6] | 32[6] | 84,213[74] | 2.0[73] | 13,145.44 |

We analyzed anatomical data from seven species, including primates (e.g., macaques) and non-primates (e.g., mice, rats, cats, tree shrews, gray squirrels, and ferrets), as detailed in Table 3. We first find that V1 RFs density (RFD) ($\rho_{\text{RF}}$) acts as a linear classifier ($y = 4.42 \times 10^4 x$), effectively distinguishing species with pinwheel structures from those with salt-and-pepper organizations. In this classifier, species like macaques, cats, tree shrews, and ferrets, which have higher RFD, are

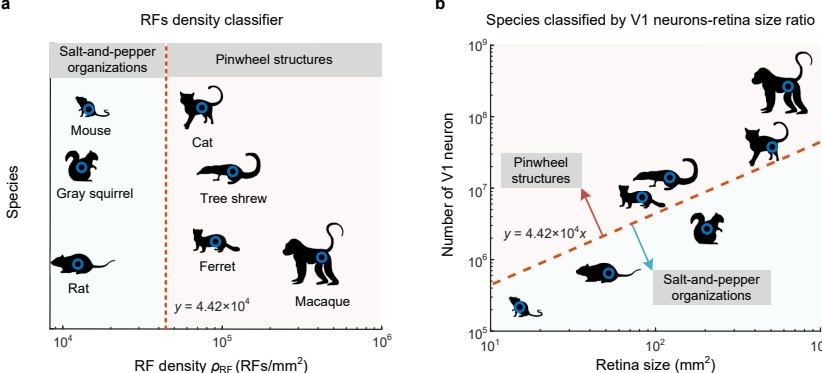

Figure 6: A linear classifier based on RFD ($y = 4.42 \times 10^4 x$) effectively differentiates species with salt-and-pepper organizations (rats, mice, gray squirrels) from those with pinwheel structures (macaques, ferrets, cats, tree shrews). **a**. This classifier reflects variations in V1 organizations across species. **b**. A plot categorizing species by the ratio of V1 neuron number to retina size acts as a divider, implying a critical ratio for the formation of pinwheel structures.

associated with pinwheel structures (light red area in Fig. 6) and exceed the classification threshold. In contrast, species with lower RFD, such as mice, rats, and gray squirrels, are linked to salt-and-pepper organizations (light blue area in Fig. 6). Thus, V1 RFD serves as a predictive metric for V1 organizational patterns across species. The $\rho_{\mathrm{RF}}$ is calculated as follows:

$$\rho_{\mathrm{RF}} = \frac{n'}{s_\mathrm{r}'} \propto \frac{n}{\left[\left(s_{\mathrm{RF}} - \varepsilon\right)\left(\sqrt{n} - 1\right) + s_{\mathrm{RF}}\right]^2}, \tag{12}$$

where $n'$ denotes the total number of neurons in V1, $s_\mathrm{r}'$ indicates the retinal surface area. We have $n' = s_{\mathrm{V1}} \times \rho_{\mathrm{V1}}$, where $s_{\mathrm{V1}}$ corresponds to the V1 2D surface area, and $\rho_{\mathrm{V1}}$ signifies the neuronal density within V1. The variable $\varepsilon$ quantifies the degree of visual input overlap among adjacent neurons, $n$ denotes the total number of neurons, and $s_{\mathrm{RF}}$ represents the RF size in the self-evolving spiking neural network (SESNN). Referring to Eq. 12 and anatomical data (Table 3), the overlap $\varepsilon$ positively correlates to V1 RFD $\rho_{\mathrm{RF}}$ and is a main factor influencing V1 RFD. We discuss the overlap as the variable of V1 organizations in the main text.

### A.4.2 SESNN reveals neuronal connection range influencing V1 clusters

The anatomical data in Table 3d for seven species show variability in V1 neuronal density ($\rho_{\mathrm{V1}}$), which influences inter-neuronal spacing and connection strength. We explore how V1 cortical orientation patterns form by adjusting the lateral connection range, impacting axon reach among E- and I-neurons, as depicted in Fig. 7. We modulate axonal arborization through parameter $\sigma$ to adjust the connection range, allowing us to simulate neuronal connections in areas with varying densities. This setup enables the SESNN model to predict changes in cortical patterns (Fig. 7). Our observations indicate that increasing axon lengths, thereby extending the connection range, enlarges hypercolumn sizes within pinwheel structures (Fig. 7d), reduces the overall number of pinwheels (Fig. 7b), and increases the nearest-neighbor pinwheel distance (NNPD) (Fig. 7c). These findings underscore the critical role of neural synaptic connection range in organizing orientation maps.

### A.5 Relationship between maximum values of local pixel entropy and local geometrical entropy for various shapes

To address the limitations of using local pixel entropy (LPE) with sliding windows alone to capture complex geometric properties, we conduct a new analysis comparing the maximum values of LPE with local geometrical entropy (LGE) across various shapes. These shapes include lines, angles, and junctions (L-, T-, X-junctions), as well as jagged edges. Both LPE and LGE values were normalized to the range [0,1] for consistency.

Let $P = \{v_1, v_2, \ldots, v_n\}$ be a polygon with vertices $v_i = (x_i, y_i)$, where $i = 1, 2, \ldots, n$. The edges of the polygon are the line segments between consecutive vertices, denoted as $e_i = \|v_{i+1} - v_i\|$,

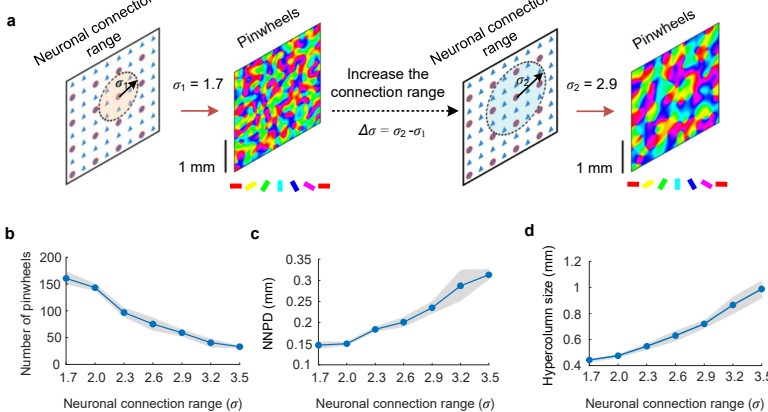

Figure 7: Neuronal connection range within V1 contributes to the formation of pinwheel structures. **a**. Modifying the synaptic connection range reshapes the dimensions of pinwheel structures. **b-d**. The relationship between the synaptic connection range ($\sigma$) and the number of pinwheels, NNPD (mm), and hypercolumn size (mm). The scale bar: 1 mm in V1 cortical surface. Color scheme: orientation preference. Lines: mean. Shaded area: SD.

where $\|\cdot\|$ represents the Euclidean distance. The angle $\theta_i$ between two consecutive edges $e_i$ and $e_{i+1}$ can be computed using the dot product:

$$\theta_i = \cos^{-1}\left(\frac{e_i \cdot e_{i+1}}{\|e_i\|\|e_{i+1}\|}\right). \tag{13}$$

With the set of edge lengths $\{e_1, e_2, \ldots, e_n\}$ and angles $\{\theta_1, \theta_2, \ldots, \theta_n\}$, we calculate the entropy for both distributions. The entropy $H$ of a discrete distribution $X$ with probability mass function $p(x)$ is given by:

$$H(X) = -\sum_{x \in X} p(x) \log p(x). \tag{14}$$

For the edge lengths and angles, the probability mass function is estimated by normalizing the frequency of occurrence of each unique edge length and angle in the polygon:

$$H(\text{Lengths}) = -\sum_{i=1}^{n} p(e_i) \log p(e_i), \tag{15}$$

$$H(\text{Angles}) = -\sum_{i=1}^{n} p(\theta_i) \log p(\theta_i). \tag{16}$$

To enhance the sensitivity of geometrical entropy to structural complexity, particularly in differentiating shapes that have similar edge lengths and angles but different structural arrangements, we introduce a scaling factor based on the logarithm of the number of vertices $n$. The defined geometrical entropy (GE) with the scaling factor is thus defined as:

$$GE = (H(\text{Lengths}) + H(\text{Angles})) \times \log(n). \tag{17}$$

This modification allows GE to capture additional complexity arising from intersections and the global arrangement of vertices, providing a more comprehensive assessment of the shape's structural intricacies.

Our results, summarized in Table 4, show that while LPE can reflect the complexity of certain patterns, it does not fully capture the geometric variations seen in more intricate shapes. For instance, the LPE values for line structures remain relatively low compared to those for jagged edges, which have the highest LPE and LGE values due to their high structural complexity. This comparison highlights the added value of incorporating LGE to better characterize local geometric structures, providing a more nuanced measure of complexity that includes both intensity distribution and spatial organization.

Table 4: Relationship between maximum values of LPE and LGE for various shapes. Both metrics are normalized to the range [0,1].

| Various shapes | Max local pixel entropy | Max local geometrical entropy |
| --- | --- | --- |
| Line 1 | 0.56 | 0.43 |
| Line 2 | 0.56 | 0.43 |
| Angle 1 | 0.81 | 0.87 |
| Angle 2 | 0.79 | 0.86 |
| Angle 3 | 0.77 | 0.87 |
| L-junction | 0.78 | 0.74 |
| T-junction | 0.78 | 0.64 |
| X-junction | 0.78 | 0.84 |
| Jagged edges | 1.00 | 1.00 |

## A.6 Pinwheel centers response to different orientation bandwidths

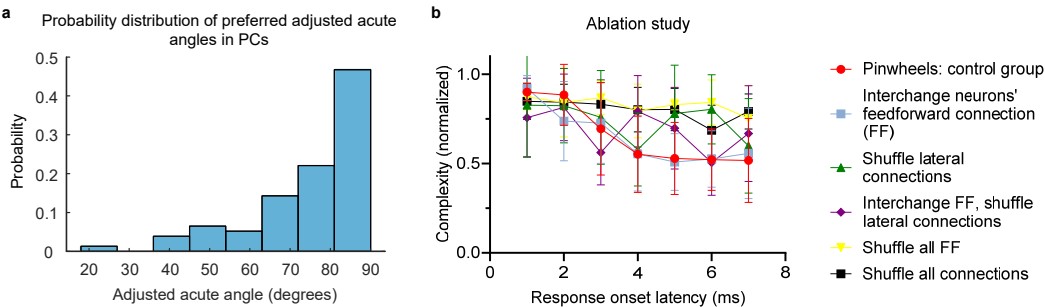

Figure 8: PCs in V1 prefer orientations and ablation study. **a**. Probability distribution of preferred acute angles in PCs. **b**. Ablation study on normalized complexity across response onset latencies. Data: mean ± SD.

Understanding the tuning of pinwheel centers (PCs) in V1 to edges, corners, and junctions is essential. In Fig. 4e, we show that PCs exhibit broader orientation tuning curves than IODs when using star-like patterns as stimuli, potentially enabling the detection of T-junctions and corners, as demonstrated by Li et al. [13] and Koch et al. [12]. We further examine the distribution of PCs' tuning curves using gratings as inputs, specifically analyzing acute angles formed by the primary and secondary peaks (Fig. 8a). This analysis reveals that PCs are more frequently associated with larger acute angles, closer to orthogonal (90°), suggesting a preference for orthogonal junctions. However, this result does not differentiate between L- and T- junctions based solely on angle. We propose that such high-order feature extraction be deferred to higher visual cortices, like V2 and V4, which are involved in texture detection, as noted by Wang et al. [79] and Roe et al. [80].

## A.7 Ablation study

We present a mechanism of multiple orientation tuning that is essential for processing complexity. Our analysis of PCs' preferred acute angles (Fig. 8a) suggests that their broad tuning enables the detection

of complex junctions, such as T- and L-junctions, likely due to variations in local connectivity within and between iso-orientation domains.

To test this, we conduct an ablation study by disrupting local connectivity and shuffling the spatial arrangement of orientation-tuned RFs in the pinwheel orientation map, while keeping other properties constant (Fig. 8b). The control group (red) maintains higher complexity over time, whereas shuffling connections—especially both feed-forward and lateral—resulted in a decline in complexity. This highlights the importance of structured connectivity in preserving complex neural responses in V1 and supports the conclusion that structured connectivity underlies enhanced saliency detection by pinwheels.

### A.8 Computing infrastructure

Table 5: Computing infrastructure

| CPU | Intel® Xeon® Gold 6348 CPU @ 2.60GHz |
|---|---|
| GPU | A100 |
| Memory | 512 GB |
| Operating system | Ubuntu 20.04.6 LTS |
| Simulation platform | MATLAB R2023a and Python 3.9 |

The simulations and analyses in this study are performed on a high-performance computing infrastructure to ensure efficient processing of large datasets and complex models. The system is powered by an Intel® Xeon® Gold 6348 CPU running at 2.60 GHz and an NVIDIA A100 GPU, providing robust computational power for intensive tasks. The system includes 512 GB of memory, which supports handling memory-intensive applications and large-scale simulations. The operating system used is Ubuntu 20.04.6 LTS, known for its stability and compatibility with scientific software. The simulations are conducted using MATLAB R2023a and Python 3.9, both of which are widely used in scientific computing and neural modeling, enabling effective implementation and analysis of the models presented in this study.

