# OpenReview forum: "Visual Pinwheel Centers Act as Geometric Saliency Detectors"
_NeurIPS.cc/2024/Conference — NeurIPS 2024 poster_

### Official Review · Reviewer_indp · 2024-07-09

**Soundness:** 3
**Presentation:** 3
**Contribution:** 2
**Rating:** 7
**Confidence:** 4

**Summary:**

This work aims to explain the origin and functional benefits of pinwheel structures in V1 compared to salt-and-pepper configurations. They use a two-dimensional self-evolving spiking neural network (SESNN) model with Hebbian-like plasticity and empirical morphological data to simulate the evolution from salt-and-pepper clusters to pinwheel structures. Their findings reveal that neurons at pinwheel centers exhibit heightened sensitivity and quicker responses to complex spatial textures in natural images, acting as primary processors for intricate contours, while adjacent iso-orientation domains refine edge representations.

**Strengths:**

- The question is of high interest: Pinwheel structures amazed experimentalist and theorist alike for decades. This is one of the main hallmarks of difference in sensory processing in primates vs rodents.

- Bioplausibility of the model: use of bioplausible modeling (spiking neurons and learning rules)

- Through citations to previous works

**Weaknesses:**

- More analyses to support the main claim in needed (see limitations)
- The use of different learning rules for E>E vs E>I is justified computationally (for stability) but not discussed in terms of bio-plausibility.

**Questions:**

- In Related works line 228, when referring to ANN models explaining pinwheels, the lack of temporal processing in these models were mentioned but their ability in connecting these maps to the overall functionality of the network is not discussed. Since the main claim of this paper is about functionality of PCs, I wonder what authors think about the functional role previous work attributed to PCs.

**Limitations:**

- In the setup, It seems that the difference between receptive filed sizes of monkeys, cats and mice was ignored. In mice receptive fields for V1 are generally much larger than monkeys (>20x, see ref below for example).

- The claim about functionality of pinwheels requires more support: the question whether the differences in responses of PCs and IODs in terms of response latency, etc have a functional role could be verified by using SESNN as a frontend for a simple DNN performing object recognition, motion detection, etc. At this point, it remains a different in response properties.

- Code is not shared at this version.

Van den Bergh G, Zhang B, Arckens L, Chino YM. Receptive-field properties of V1 and V2 neurons in mice and macaque monkeys. J Comp Neurol. 2010 Jun 1;518(11):2051-70. doi: 10.1002/cne.22321. PMID: 20394058; PMCID: PMC2881339.

---

> ### Author Rebuttal · Authors · 2024-08-07
>
> Thank you for your detailed review and insightful comments on our manuscript.
> ## Points Raised:
> ### 1. Bioplausibility of Learning Rules for E>E vs E>I Connections:
> **Response:** We recognize the importance of bioplausibility in modeling neuronal networks. In the context of our SESNN model, biological plausibility in learning rules for E>E and E>I connections is essential for simulating realistic neural behavior. We integrate principles of Hebbian-like plasticity, similar to biological brains, where synaptic connections between neurons strengthen based on their co-activation. E>E connections, according to [1], is set to be weaker than E>I connections. By incorporating these bioplausible learning rules, SESNN models can better balance excitation and inhibition that is crucial for maintaining stable neural activity levels and emulate the complex processing capabilities observed in biological neural networks.
> 1. Hofer, S. B., Ko, H., Pichler, B., Vogelstein, J., Ros, H., Zeng, H., … & Mrsic-Flogel, T. D. (2011). Differential connectivity and response dynamics of excitatory and inhibitory neurons in visual cortex. Nature Neuroscience, 14(8), 1045-1052.
> ### 2. Functional Role of Pinwheel Centers in Previous Work:
> **Response:** The three cited works provide significant insights into the functional roles attributed to pinwheel centers (PCs) in the visual cortex from different perspectives. One study explores how the retino-cortical mapping ratio influences the organization of visual cortical maps, including columnar structures and salt-and-pepper patterns[1]. Another research[2] investigates how orthogonal tiling projections from the retina to the visual cortex contribute to the formation of visual maps, including the arrangement of pinwheel centers. Also, a study[3] presents a network model that explains the emergence of simple and complex cells in the visual cortex, with PCs playing a pivotal role in this development. Though they haven't consider PCs latency as salieny detector, the three cited works collectively highlighted the multifaceted roles of pinwheel centers in visual processing.
> 1. Jaeson Jang, Min Song, and Se-Bum Paik. Retino-Cortical Mapping Ratio Predicts Columnar and Salt-and-Pepper Organization in Mammalian Visual Cortex. *Cell Reports, 30*(10), 3270-3279.e3, March 2020.
> 2. Min Song, Jaeson Jang, Gwangsu Kim, and Se-Bum Paik. Projection of Orthogonal Tiling from the Retina to the Visual Cortex. *Cell Reports, 34*(1), January 2021.
> 3. Louis Tao, Michael Shelley, David McLaughlin, and Robert Shapley. An egalitarian network model for the emergence of simple and complex cells in visual cortex. *Proceedings of the National Academy of Sciences, 101*(1), 366-371, January 2004.
> ### 3. Receptive Field Sizes Across Species:
> **Response:** In our study, we acknowledge the differences in receptive field sizes across species such as monkeys, cats, and mice. The focus of our research is primarily on the overlap and interaction within orientation maps, where we have normalized receptive fields for comparative analysis. The size of receptive field in v1 is more about resolution than orientation map formation, for example theres more than ten times change in the rf size from fovea to peripheral in macaque v1 while very little change in the properties of the corresponding orientation map[1,2]. Therefore we expect little effects from receptive field shift sizes across species as long as the overlap is constant. However, if time permits, we can further verify this in our model during the discussion period.
> 1.  Bosking, W. H., Zhang, Y., Schofield, B., & Fitzpatrick, D. (1997). Orientation selectivity and the arrangement of horizontal connections in tree shrew striate cortex. The Journal of Neuroscience, 17(6), 2112-2127.
> 2.  Horton, J. C., & Hocking, D. R. (1996). Intrinsic variability of ocular dominance column periodicity in normal macaque monkeys. Journal of Neuroscience, 16(22), 7228-7339.
> ### 4. Verification of Functional Claims with Simple DNN:
> **Response:** We designed a Spiking Neural Network (SNN) for classifying the Fashion MNIST (FMNIST) dataset. The images were first processed by the SESNN network, generating a 20ms spike train. This spike train was then fed into a convolutional SNN. The final layers consisted of a flattened layer, followed by two linear layers, leading to the classification output. The network was trained using surrogate gradient methods. For comparison, we also evaluated a version of the SESNN where the neuronal activity was resampled based on firing rates to generate Poisson spike trains, effectively removing latency.
> The classification accuracy of the SNN with and without latency across different classes of the FMNIST dataset is summarized in the tables below:
> |Dataset Class| Accuracy(%) of Model With Latency| Accuracy(%) of Model Without Latency|
> |-|-|-|
> | T-shirt / top| 86.80 | 85.40|
> |Trouser| 97.30|96.40|
> |Pullover| 83.40| 83.80|
> |Dress| 91.10| 90.30|
> |Coat| 83.60| 81.60|
> |Sandal|96.20|96.80|
> |Shirt|73.20|67.80|
> |Sneaker| 95.40| 95.40|
> |Bag| 97.80| 97.00|
> |Ankle boot|96.10|94.20|
> |**Overall**|**90.09**|**88.87**|
> This experiment elucidates the functional significance of differences in response latency between pinwheel centers and iso-orientation domains (IODs). We also did decoding and reconstruction part with results addressed in Fig.S(b) of Supplemental PDF File.
> ### 5. Sharing of Code:
> **Response:** The code is shared at https://github.com/HenryGithub1?tab=following.
> ## Conclusion:
> Your feedback has been invaluable in identifying areas for enhancement. We appreciate the opportunity to improve our paper and are committed to delivering a revised version that meets the standards of thorough evaluation and clarity. Thus, we kindly urge you to reconsider the score in light of our clarifications. We remain available to provide further explanations if there are any additional questions. Thank you once again for your thorough review and constructive input.

---

> > ### Comment · Reviewer_indp · 2024-08-10
> >
> > Thank you for clarifications. I raised the score.

---

> > > ### Author Response · Authors · 2024-08-13
> > >
> > > Thank you for your interest in our work. We appreciate you taking the time to review it.

---

### Official Review · Reviewer_FA4k · 2024-07-11

**Soundness:** 3
**Presentation:** 4
**Contribution:** 3
**Rating:** 6
**Confidence:** 3

**Summary:**

This paper introduced a novel spiking neural network (SNN) to investigate the functional roles of pinwheel structures in the primary visual cortex of higher mammals and primates. By adjusting a visual RF overlapping parameter, their model can produce the salt-and-pepper and pinwheel organizations observed in lower (rodent) and higher (macaque and cat) mammals. Their results suggest neurons in pinwheel centers are more responsive towards complex geometry and spatial textures than those in the iso-orientation domains.

**Strengths:**

- This paper investigates the important problem of salt-and-pepper vs pinwheel structure observed in the mammalian visual cortex. Their model shows that visual overlap can influence the topological organization in V1 is interesting, and that the resulting orientation preference maps match experiential data obtained from rodents, macaques, and cats.
- The paper is very well-written. Moreover, analysis of the spatial-temporal response pattern and response time with respect to the complexity of the visual scene is novel and interesting.

**Weaknesses:**

- The paper lacks a comparison with previous work or verification with experimental data. For instance, only one metric from the baseline model is reported in Table 1. Please see points 2 and 3 in “Questions”.
- The SESNN model shows a rippling effect/pattern (Figure 2) and a heightened response with high contour complexity (Figure 3) in pinwheel centers, it is unclear whether or not this is indeed the case in the visual cortex without validation/comparison with experimental data.

**Questions:**

- One of the main claims is the differences in neuronal response time with respect to the complexity of the visual scene between salt-and-pepper and pinwheel organizations. Based on Figure 3c, other than the initial time point (1 ms), the two structures seem to have similar response ranges and a downward trend as latency increases. Can the author comment on that?
- In Table 1, the authors compare a baseline model and the SESNN model against the experimental data obtained from macaques. Can the authors clarify why the NNPD and hypercolumn size are “N/A” for the baseline model? Without a proper baseline, it is difficult to evaluate how well the proposed model is performing in this specific task.
  - Moreover, can the authors clarify why the pinwheel density is omitted as a metric in Section 2.1, though it is included in Table 1?
  - Finally, I believe a brief description of the baseline model, and how it differs from SESNN, should be added to the main text.
- I believe the analysis would be more complete if the authors could add the pinwheel data from rodents and cats in Table 1 to compare the performance of the proposed model against the baseline.

**Limitations:**

I strongly suggest the authors discuss the limitations of this work and potential future research direction in the paper.

---

> ### Author Rebuttal · Authors · 2024-08-07
>
> # Rebuttal for Reviewer FA4k
> Thank you for your thorough review and valuable feedback on our paper. We appreciate your recognition of the strengths of our model and its presentation.
> ## Points Raised:
> ### 1. Neuronal Response Time to Complexity of Visual Scene:
> **Response:** Based on our findings, pinwheel centers (PCs) exhibits quicker and stronger neuronal responses to complex spatial textures in natural visual scenes. This contrasts with neurons organized in salt-and-pepper configurations, which generally exhibit slower responses to such stimuli. Though the downward trends look similar in both structures the slope is higher for pinwheels and demonstrate a clear preference for more complex inputs by responding quicker.
>
> ### 2. Comparison with Previous Model:
> **Question1:** I believe a brief description of the baseline model, and how it differs from SESNN, should be added to the main text.
>
> **Response:**  We investigated previous model and found that the "baseline" model's performance in generating a complete pinwheel structure, crucial for validation by our metrics, did not meet our criteria, leaving NNPD and hypercolumn size not available for calculation. Therefore, "N/A" was appropriately indicated in Table 1. However we may have misused the word "baseline", "a previous model" is more appropriate.
> Specificaly, the mentioned model[1] is a spiking neural network model simulating the formation of orientation and ocular dominance maps in the visual cortex. In contrast, the SESNN model in the current study not only forms orientation and ocular dominance maps but also involve both salt-and-pepper clusters and pinwheel structures. SESNN overtakes the baseline model in generation of Pinwheel structures, and shows that Pinwheel centers act as first-order processors with heightened sensitivity and reduced latency to intricate contours. This advancement underscores the SESNN model’s ability to better mimic the functional advantages observed in the visual cortex of higher mammals, providing a significant improvement over previous approaches.
> 1. Srinivasa, N., & Jiang, Q. (2013). Stable learning of functional maps in self-organizing spiking neural networks with continuous synaptic plasticity. *Frontiers in Computational Neuroscience, 7*, 10.
>
> **Question2:** Can the authors clarify why the pinwheel density is omitted as a metric in Section 2.1, though included in Table 1?
>
> **Response:** In a paper published in 2010[1], it was established that pinwheel density is approximately 3.14, which is considered a reliable metric for assessing the quality of pinwheel structures. Our model also measured pinwheel density, consistently finding it around three, aligning well with experimental evidence. In Figure 2, we varied the overlap size, and given our earlier analysis, the density remains around three in our model with reasonable change in overlap (as long as it does not cause a transition to salt-and-pepper structure). Therefore, it wasn't necessary to use pinwheel density to measure the impact of overlap. However, we included pinwheel density in Table 1 because it is a crucial benchmark for comparing the quality of pinwheel structures across different models, including our baseline model and real animal data.
> 1. Kaschube, M., Schnabel, M., Löwel, S., Coppola, D. M., White, L. E., & Wolf, F. (2010). Universality in the Evolution of Orientation Columns in the Visual Cortex. Science, 330(6007), 1113-1116. doi:10.1126/science.1194869​
>
> ### 3.  Include Experimental Data to Compare:
> **Response:**
> We acknowledge the importance of experimental data and did extra investigation on the experimental data from rodents and cats[1-3] and will add into the revised manuscript. This addition will provide a broader comparative analysis and strengthen the validation of our model across different mammalian species. Yet we didn't identify existing research that can explicitly address our conclusion. However, this is the novelty point of our work from the perspective of computational models, we'll try to launch behavioral experiment evidence soon by cooperating with experimentalists.
> 1. Stryker, M. P., Sherk, H., Leventhal, A. G., & Hirsch, H. V. Physiological consequences for the cat's visual cortex of effectively restricting early visual experience with oriented contours. Journal of Neurophysiology 1978, 41(4), 896-909.
> 2. Tanaka, S., Miyashita, M., Wakabayashi, N., O’Hashi, K., Tani, T., & Ribot, J. (2020). Development and reorganization of orientation representation in the cat visual cortex: Experience-dependent synaptic rewiring in early life. Frontiers in Neuroinformatics, 14, Article 41.
> 3. Vita, D. J., Orsi, F. S., Stanko, N. G., et al. Development and organization of the retinal orientation selectivity map. Nat Commun 2024, 15, 4829.
> ### 4. Discussion of Limitations and Future Work
> **Response:** We recognize the need to discuss the limitations of our work more comprehensively and outline potential future research directions in the paper. This will ensure transparency and guide further advancements in this area of study. Several limitations should be considered, including potential oversimplification of biological neural networks, and the exclusive focus on functions of spatial aspects of the saliency map. Future research could focus on empirically validating these findings using neurophysiological techniques in biological models, developing dynamic computational models that can address more apects of saliency comprehensively to broaden our understanding of neural network functionality and advance applications in neuroscience and artificial intelligence.
> ## Conclusion:
> Your feedback has been instrumental. By enhancing our comparisons with previous work, providing further validation with experimental data, and addressing the noted limitations, we aim to strengthen the impact and robustness of our study. Thank you once again for your valuable feedback and for the opportunity to improve our paper.

---

> > ### Comment · Reviewer_FA4k · 2024-08-13
> > **Response to rebuttal**
> >
> > I thank the authors for their thorough responses and they have addressed most of my main concerns. I have therefore updated the score.
> >
> > As to "Neuronal Response Time to Complexity of Visual Scene”, it is not visually obvious that the response time downward trend is stronger in the pinwheel structure. I suggest the authors quantify the downward trends in Figure 3c between pinwheels vs salt-and-pepper, and show their statistical significance at all time points.

---

> ### Author Response · Authors · 2024-08-13
>
> Thank you for acknowledging our effort. As for quantifying the difference between downward trends in pinwheels and salt-and-pepper, we consider statistical significance between every other 2 ms for both pinwheel and salt-and-pepper structures to show the downward trend is only significant among pinwheels. (Since recorded neurons are not guaranteed to be in the same pinwheel, the statistical significance of a direct comparison of slopes is unavailable.):
> | Latency-Gap(ms) | p for Pinwheel | p for Salt & Pepper|
> |-------------|---|---|
> |1-3  |**0.00002**|0.25352|
> |3-5  |**0.01208**|0.73593|
> |5-6  |0.64746|0.16130|
> |7-9  |0.95521|0.99757|
>
> The statistical significance of pinwheels vs salt-and-pepper at all time points:
> | Latency(ms) | 1 | 2 | 3 | 4 | 5 | 6| 7| 8| 9| 10|
> |-------------|---|---|---|---|---|---|---|--|--|---|
> | p (Pinwheel vs Salt & Pepper)    |**0.0001**|**0.0180**|**0.0034**|**0.0173**|0.2146|0.1233|**0.0490**|**0.0004**|**0.0274**|0.5049|

---

### Official Review · Reviewer_Boae · 2024-07-11

**Soundness:** 3
**Presentation:** 3
**Contribution:** 2
**Rating:** 6
**Confidence:** 5

**Summary:**

The authors present a comprehensive model of the primary visual cortex adapted for various mammalian species, demonstrating its ability to reproduce orientation maps and compatibility with experimental data across different factors such as neuron density or RF overlap. Importantly, they provide evidence in their model that pinwheel centers act as saliency detectors.

**Strengths:**

While the result put forward by the paper may seem intuitive, as a PC contains in a close neighborhood cells selective to different orientations, and thus may be a good candidate for a saliency detector, yet this model gives some interesting quantitative predictions.

**Weaknesses:**

Given the link with neuroscience, more links with behavioural data would be beneficial: are animals without PCs less efficient in detecting saliency? Is the density of PCs compatible with the "resolution" of the saliency map? In general, this feature of PCs as detecting features containing multiple orientations should be more broadly tested, for instance by using existing results on textures with different orientation bandwidths. such work could help better undertand the underlying principles which give rise to that particular feature.  (The link to https://github.com/HenryGithub1?tab=following (and to its followers) may reveal the authorship, which may be a problem for NeurIPS. Use anonymous links instead. There is a strong overlap with submission 16614, most certainly from the same authors.)

**Questions:**

In the computation of entropy, why use luminance values instead of that of the (whitened) images used during the training? Also the numbers used in the paper and synthesized in table 2 are given following previous papers, bt it is not said at what eccentricity they are computed. Could you please provide this information? How do you justify that the emergence is uniform while retinotopic space is not (certainly thanks to the fact that RF size is function of eccentricity, as well as the RF density, but this is not discussed in the paper as far as I could read)?

**Limitations:**

The main result is figure 3, and to better the mechanisms leading to this result, an ablation study would be an asset to the paper. Minor: in Figure 4c, your error bar escapes the imit of valid values (normalized entropy higher than 1). Use quantile regression to estimate the 95% confidence interval?

---

> ### Author Rebuttal · Authors · 2024-08-07
>
> # Rebuttal for Reviewer Boae
> Thank you for your insightful feedback and constructive comments on our paper.
> ## Points Raised:
> ### 1. Behavioral Data and Saliency Detection:
> **Question1:** Are animals without PCs less efficient in detecting saliency?"
>
> **Response:** We acknowledge the importance of behavioral data, yet we didn't identify existing research that can explicitly prove with behavioral data that animals lacking well-defined pinwheel centers (PCs) in their visual cortex may indeed be less efficient in detecting saliency compared to species with distinct pinwheel organizations. However, we do expect so as PCs are critical for integrating information from surrounding iso-orientation domains (IODs), allowing for enhanced sensitivity to complex visual features like edges and textures[1]. Since most related experiments didn‘t focus on this address, which is the novelty point of our work from the perspective of computational models, we'll try to launch behavioral experiment evidence soon by cooperating with experimentalists.
>
> **Question2:** Is the density of PCs compatible with the 'resolution' of the saliency map?
>
> **Response:** If we understand correctly, the 'resolution' of the saliency map should be referring to resolution of image. Then, the density of PCs within the cortex correlates with the resolution of the saliency map, where higher densities typically enable finer discrimination of visual stimuli. In general, comparative neuroanatomy studies across species have shown that variations in PC density and organization impact visual processing capabilities, influencing the ability to detect and respond to salient visual cues[2].
>
> 1. Bosking, W. H., Zhang, Y., Schofield, B., & Fitzpatrick, D. (1997). Orientation selectivity and the arrangement of horizontal connections in tree shrew striate cortex. The Journal of Neuroscience, 17(6), 2112-2127.
> 2. Angelucci, A., & Rosa, M. G. (2015). Resolving the organization of the third tier visual cortex in primates: A hypothesis-based approach. Visual Neuroscience, 32(E010).
> ### 2. Testing on Textures with Different Orientation Bandwidths:
> **Response:** We agree that such work could help better undertand the underlying principles which give rise to that particular feature and will incorporate results from textures with different orientation bandwidths to validate our model further. This will help clarify the underlying principles governing the observed features in PCs.
>
> ### 3. Computation of Entropy using luminance values:
> **Response:** With our definition of saliency, we want to control other saliency components like contrast, and use binarized luminance value to isolate the effect of geometric complexity.
> ### 4. Uniform Emergence and Retinotopic Space:
> **Question1:** "The numbers used in the paper and synthesized in table 2 are given following previous papers, but it is not said at what eccentricity they are computed. Could you please provide this information?"
>
> **Response:** For mouse[1] and cat[2], the receptive field eccentricity ranged from 0 to 90°, and the corresponding level used here is around 0°. For macaque[3], the eccentricity ranges from 0 to 15°. Here we all used data corresponding to lowest eccenctricity level at 0°.
> 1. van Beest, E.H., Mukherjee, S., Kirchberger, L. et al. Mouse visual cortex contains a region of enhanced spatial resolution. Nat Commun 12, 4029 (2021).
> 2. Wilson, J. R., & Sherman, S. M. (1976). Receptive-field characteristics of neurons in cat striate cortex: changes with visual field eccentricity. Journal of neurophysiology, 39(3), 512-533.
> 3. Tehovnik, E. J., & Slocum, W. M. (2007). Phosphene induction by microstimulation of macaque V1. Brain Research Reviews, 53(2), 337–343.
> **Question2:** How do you justify that the emergence is uniform while retinotopic space is not?
>
> **Response:** The uniform emergence of orientation maps in the visual cortex, despite non-uniform retinotopic space, underscores the sophisticated self-organizing principles governing neural development, and is primarily facilitated by receptive field (RF) size with eccentricity and the density of RFs across the visual field[1]. Neurons in the visual cortex adaptively adjust their RF sizes according to their distance from the fovea, ensuring optimal sensitivity to visual features at different spatial resolutions[2]. Additionally, the density of RFs compensates for variations in retinal input, with higher eccentricity regions exhibiting smaller, densely packed RFs[3].
> ### 5. Limitations:
> **Question1:** Ablation study.
>
> **Response:** We ablate the local connectivity of trained pinwheel orientation map while keeping other properties. This result indicate the structured connectivity from pinwheel may be the underlying mechanism for its better saliency. Due to time limit we're not able to conduct more detailed ablation studies, however we thank your suggestions and will furnish our word with more ablation studies soon.
>
> **Question2:** Minor correction in Figure 4c.
>
> **Response:** Thanks for the suggestion, to address this question we now use boxplot to show the trend and statistical features as provided in  supplemental PDF file.
>
> ### 6. Anonymous Links and Overlap with Submission:
> **Response:** We will replace the GitHub link with anonymous references and clearly distinguish any shared content with submission 16614 through proper citation and differentiation.
>
> ## Conclusion:
> Your feedback has been invaluable in identifying areas for enhancement. Thank you once again for your thorough review and constructive suggestions. We are committed to refining our work to contribute effectively to the field of visual processing and the role of pinwheel centers.

---

> > ### Author Response · Authors · 2024-08-14
> >
> > We are currently awaiting feedback on our rebuttal from Reviewer Boae. We have thoroughly addressed all the concerns and eagerly anticipate their insights, which would significantly contribute to enhancing our work.

---

> ### Author Response · Authors · 2024-08-08
>
> We are now providing this additional information to ensure a comprehensive understanding of our study and to address your valuable feedback more thoroughly.
>
> **About behavioral data**: Linking our model with behavioral data could further strengthen our findings. Currently, no studies show that animals (like rodents) without PCs are less efficient in detecting saliency. However, electrophysiological studies suggest that PCs have delayed response latency, indicative of higher-order processing [citations 6, 7, 21 in our paper]. This arises from using drifting grating stimuli that activate IODs more readily. Koch et al. [citation 6 in our paper] note that IODs exhibit cross-orientation suppression under complex stimuli, narrowing their tuning, unlike PCs. This study, however, omits temporal neural data within pinwheel structures. The SESNN model aligns with physiological findings, showing that IODs and PCs favor single and complex orientation stimuli, respectively.
>
> **About the textures with different orientation bandwidths**: Understanding the tuning of PCs in V1 to various edges, corners, and junctions is crucial. In Fig. 4e of our paper, we show that PCs exhibit broader orientation tuning curves than iso-orientation domains, which may allow them to detect T-junctions and corners, as demonstrated by Ming Li et al. (2019, *Science Advances*) and Erin Koch et al. (2016, *Nature Communications*). Your insightful comment prompted us to further examine the distribution of PCs' tuning curves. We have analyzed the acute angles formed by the primary and secondary peaks in the orientation tuning curve (**Table 1**). **Table 1** shows that PCs are more frequently associated with larger acute angles (closer to orthogonal, 90°), indicating a preference for orthogonal junctions. However, this result does not distinguish between "L" and "T" junctions beyond their angle. We suggest deferring such high-order feature extraction to higher visual cortices like V2 and V4, which are involved in texture detection, as discussed by Tianye Wang et al. (2024, *Nature Communications*) and Anna W. Roe et al. (2012, *Neuron*).
>
> **Table 1: Probability distribution of preferred adjusted acute angles in pinwheel centers.** (Corresponding figure is available in anonymous GitHub repository)
>
> | Adjusted acute angle range (degrees) | Probability (%) |
> |-----------------------|-----------------|
> | 0 - 9                 | 0               |
> | 9 - 18                | 0               |
> | 18 - 27               | 1.30            |
> | 27 - 36               | 0               |
> | 36 - 45               | 3.90            |
> | 45 - 54               | 6.49            |
> | 54 - 63               | 5.19            |
> | 63 - 72               | 14.29           |
> | 72 - 81               | 22.08           |
> | 81 - 90               | 46.75           |
>
> **Ablation study**: In our paper (Fig. 4e), we present a mechanism of multiple orientation tuning that is crucial for processing complexity. Our experiment (**Table 1**) on PCs' preferred acute angles suggests that their broad tuning enables detection of complex junctions like T- and L-junctions, likely due to differences in local connectivity within and between iso-orientation domains.
>
> We appreciate the reviewer's feedback and conducted an ablation study by disrupting local connectivity and shuffling the spatial positions of orientation-tuning receptive fields in the pinwheel orientation map, while keeping other properties constant (**Table 2**). This supports the conclusion that structured connectivity, as shown in Fig. 4e, underlies the enhanced saliency detection by pinwheels.
>
> **Table 2: Ablation study on local connectivity and orientation-tuning receptive fields (available in anonymous Github). The table shows normalized entropy values (mean ± std).**
>
> | Latency (ms) | 1| 2| 3| 4| 5| 6| 7|
> |-|-|-|-|-|-|-|-|
> |Pinwheels (control group) | 0.915 ± 0.049 | 0.950 ± 0.170 | 0.899 ± 0.259 | 0.468 ± 0.214 | 0.462 ± 0.202 | 0.491 ± 0.173 | 0.491 ± 0.235 |
> | Interchange feedforward connection (FF)| 0.948 ± 0.070 | 0.897 ± 0.222 | 0.889 ± 0.226 | 0.558 ± 0.208 | 0.503 ± 0.159 | 0.486 ± 0.158 | 0.464 ± 0.253 |
> |Shuffle lateral connections| 0.959 ± 0.288 | 0.916 ± 0.208 | 0.909 ± 0.262 | 0.500 ± 0.205 | 0.947 ± 0.270 | 0.874 ± 0.194 | 0.501 ± 0.264 |
> |Interchange FF, shuffle lateral connections| 0.866 ± 0.221 | 0.904 ± 0.186 | 0.524 ± 0.181 | 0.907 ± 0.200 | 0.626 ± 0.229 | 0.492 ± 0.184 | 0.738 ± 0.268 |
> |Shuffle all FF| 0.893 ± 0.073 | 0.937 ± 0.193 | 0.816 ± 0.093 | 0.783 ± 0.151 | 0.820 ± 0.029 | 0.905 ± 0.126 | 0.837 ± 0.184 |
> |Shuffle all connections| 0.849 ± 0.101 | 0.875 ± 0.100 | 0.888 ± 0.136 | 0.821 ± 0.125 | 0.818 ± 0.119 | 0.699 ± 0.140 | 0.767 ± 0.098 |

---

> > ### Comment · Area_Chair_KHne · 2024-08-08
> >
> > Please note that you should not do this, as was explicitly pointed out by the instructions sent out yesterday:
> >
> > > The deadline for submitting the rebuttal has now passed. The reviewers will now read the rebuttals you posted. When relevant, they will ask for clarification questions. **The discussion phase is meant to clarify these questions, rather than to provide further comments regarding the reviews.**
> >
> > I will leave it to the reviewers to decide whether they want to take this comment into account or not.

---

> > > ### Author Response · Authors · 2024-08-14
> > >
> > > Thank you for the clarification. I apologize for any misunderstanding regarding the submission guidelines. Our intention was to help the reviewers better understand our work.

---

### Official Review · Reviewer_TFr2 · 2024-07-15

**Soundness:** 2
**Presentation:** 2
**Contribution:** 2
**Rating:** 4
**Confidence:** 4

**Summary:**

This paper uses a self-evolving spiking neural network model to investigate why some visual systems develop pinwheel structures while others have salt-and-pepper organization of orientation tuning. The simulation shows that the organization depends on the amount of receptive field overlap between neighbouring neurons. Experiments on the trained network show that pinwheel centers respond more efficiently to complex edge structure in images than salt-and-pepper neurons.

**Strengths:**

+The findings relating orientation tuning organization to receptive field overlap are quite interesting and seem novel.
+The modelling approach seems well-designed and has a lot of potential to investigate the evolutionary advantages of different types of organization for different images/tasks.

**Weaknesses:**

- The experiments seem to use binary edge map images only, if I’ve understood them correctly. No visual system evolved to process these kinds of binary images, so it’s unclear what conclusions we should draw from these experiments. It would be much more useful to show experimental results for real images.
- Entropy in a local region of the edge map is used as a proxy for geometric complexity, but this just measures the proportion of pixels in a local part of the edge map which are white vs. black.  A little patch of random noise (50% white and 50% black pixels) would have maximum entropy but no geometric information. From the experiments, it’s hard to tell what aspect of the high-entropy regions the pinwheel centers are more responsive to – would they respond highly to random noise? Or are they responding to edge intersections? Corners? Textures? Given that the input to the model is a binary edge map, there are many ways the edges could be characterized to better understand the tuning.

**Questions:**

Do the pinwheel centers show faster responses to regions of the image that are ground-truth boundary vs. other regions of the image? (If a BSDS image is input to the model, do the pinwheel centers respond differently to the pixels which are 1 in the edge map than those which are 0?) This might show an advantage for these cells in boundary detection or figure-ground analysis.

Are pinwheel centers tuned for particular types of edge structure (for example, giving a different response for T-junctions than other types of junctions)?

---

> ### Author Rebuttal · Authors · 2024-08-07
>
> # Rebuttal for Reviewer TFr2
> Thank you for your detailed feedback on our paper. We appreciate your recognition of the novelty in our approach and the potential of our modeling technique.
> ## Points Raised:
> ### 1. Use of Binary Edge Maps:
> **Response:** We indeed used whitened natural images to train the model and tested with binary image map. We used binary edge maps as an initial test to isolate the geometric complexity of natural images while eliminating the influence of luminance and contrast variations. By using binary images, we ensure that the model focuses on detecting complex features inherent in natural scenes, rather than being influenced by luminance differences and thus explicitly demonstrate that pinwheel centers respond more efficiently to complex edge/contour structures. Luminance and contrast are important elements for saliency, complementing complexity saliency. In this case, we will further consider natural images to validate our model's performance in more realistic scenarios. Future experiments will involve testing with real-world images to validate our model's performance in more realistic scenarios, particularly in boundary detection and figure-ground separation. As addressed in Fig.S1(a) in supplemental PDF, we indeed observe PC's significantly shorter latency in response to edges than other regions of the BSDS500 natural images.
> ### 2. Entropy as a Proxy for Geometric Complexity:
> **Response:** The concept of using entropy in a local region of the edge map as a proxy for geometric complexity can be encapsulated by the following equation which is derived from the idea of Shannon's mutual information[1]:$$H = H_{Geometric\ Complexity} + H_{Noise}$$
> This equation suggests that the entropy of an image region is composed of both geometric information and noise. In our current approach, we hypothesize that the noise level in the edge maps (artificial star-like binary images and BSDS 500 groundtruth in Figs. 3 and 4) is negligible, allowing us to consider entropy as a direct measure of geometric complexity. By assuming that noise is zero, we can simplify our analysis and use entropy to reflect the structural information within the image accurately.
> However, we acknowledge the potential influence of noise on entropy measurements. To address this, we plan to include a noise term in our study soon, which will help us differentiate between random noise and true geometric complexity, ensuring a more robust and accurate assessment of image structure.
> 1. Shannon, C. E. (1948). “A Mathematical Theory of Communication.” Bell System Technical Journal, 27(3), 379-423.
> ### 3. Pinwheel Centers' Response to Ground-Truth Boundaries and Specific Edge Structures:
> **Question1:** Do the pinwheel centers show faster responses to regions of the image that are ground-truth boundary vs. other regions of the image?
> **Response:** Our model demonstrates that pinwheel centers respond faster to binarized boundaries, indicating their role as saliency detectors without the influence of other elements. We consider additional tests using diverse datasets to provide a comprehensive evaluation of our model's functionality toward edge detection in non-binarized original images, and indeed observed statically significant speed advantage of PC in detection of edges (>5 ms) than other regions (~10ms) as illustrated in Fig.S1(a), Supplemental PDF file.
> **Question2:** Are pinwheel centers tuned for particular types of edge structure?
> **Response:** Understanding the tuning of pinwheel centers to various edge junctions and textures is crucial. Pinwheel centers have the broadest orientation tuning curves, as verified by Fig. 4e. They significantly contribute to detecting T-junctions and corners, as shown by citation 7 in our paper[1]. Additionally, textures are detected in V4, as demonstrated by Tianye Wang et al. in their 2024 Nature Communications article. Due to time limitations, we will provide the new results during our discussion period. Thank you for understanding.
> 1.Wang, T., Lee, T.S., Yao, H. et al. Large-scale calcium imaging reveals a systematic V4 map for encoding natural scenes. Nat Commun 15, 6401 (2024).
> ## Conclusion:
> While we acknowledge the limitations highlighted in the review, we believe that our initial findings provide a strong foundation for future research. We appreciate the opportunity to improve our work and are confident that these enhancements will demonstrate the robustness and applicability of our model in understanding visual processing.  We have made every effort to address the Reviewer TFr2's concerns and kindly urge you to reconsider the score in light of our clarifications. We remain available to provide further explanations if there are any additional questions. Thank you for your valuable feedback.

---

> > ### Comment · Reviewer_TFr2 · 2024-08-08
> > **Response to rebuttal**
> >
> > I understand how the entropy measure is calculated, but it doesn't seem to distinguish between increasing geometric complexity (e.g., more highly-branching junctions) and increasing texture (e.g., more jagged-looking edges). The measure only cares about how many of the pixels in the window are white. The entropy of a straight line and a corner are the same according to this measure, even though the corner is (I assume?) more geometrically complex. It seems like in most cases, the highest-entropy parts of the image would be textured natural background structures like trees or mountains, while the more important foreground objects (like a person) may have lower entropy because they have smoother boundaries.
> >
> > I also don't really understand what we can take away from the model's response to binary edge maps, given that the cells were trained on natural images. The analysis in Fig.S1(a) looks promising. What is the pattern for salt-and-pepper?
> >
> > Also, just out of curiousity, why are the images filtered as shown in the supplemental? It looks like they have been bandpass filtered to remove low spatial frequency information.

---

> > > ### Author Response · Authors · 2024-08-14
> > >
> > > We are still awaiting feedback on our rebuttal from Reviewer TFr2. We have carefully addressed all the concerns and sincerely hope to receive their insights soon, as it would greatly assist us in improving our work. If our clarifications address your concerns satisfactorily, we would greatly appreciate it if you could consider raising the score.

---

> ### Author Response · Authors · 2024-08-12
> **Response to Official Comment 1**
>
> Thank you for your insightful comments regarding the distinction between geometric complexity and texture complexity. We appreciate the opportunity to clarify our approach and address your concerns:
>
> **How to distinguish geometric complexity and texture**:
>
> 1. **Local pixel entropy with sliding windows (LPESW)**: Global pixel entropy, indeed, does not measure geometric complexity within a window. Instead, we used MATLAB's `entropyfilt` function, which calculates local entropy for each pixel's neighborhood using sliding windows. This LPESW method evaluates the variation and complexity of pixel spatial distributions. While it doesn’t directly measure geometric shapes, it reflects their complexity through local intensity variations. By sliding a small window across the binary images, the local entropy measure captures the spatial distribution of entropy within these windows. This spatial distribution inherently includes geometric information, allowing it to detect local changes in intensity that characterize edges, corners, and other complex patterns.
> 2. **Verification with additional experiments by LPESW**: To verify our paper's approach, we conducted new experiments using various shapes, including lines, angles, L-, T-, and X-junctions (geometric complexity), as well as jagged edges (texture). The maximum entropy values obtained were 0.52 (lines), 0.72 (angles), 0.73 (junctions), and 0.94 bits (jagged edges), respectively. These results confirm that our LPESW method is sensitive to the complexity of geometric structures. (Available in the previously mentioned anonymous GitHub repository.)
> 3. **Comparison with geometrical entropy (GE)**: To further validate LPESW, we first approximate the contours of the shape using the Ramer-Douglas-Peucker algorithm, which simplifies the contour by reducing the number of vertices while preserving the general shape. The resulting vertices form a polygon, which serves as the basis for calculating the distribution of edge lengths and angles. GE is then defined as the sum of entropy values from these two distributions. We have tested the same shapes (as mentioned in point 2) using the GE measure. The results showed a strong correlation with in max values with LPESW (**table 1**) when calcuated with siliding windows, confirming the validity of LPESW for our experiments. (Also available in the previously mentioned anonymous GitHub repository.)
> 4. **Recognition of white noise**:  While the global GE for the pattern returns a NaN directly for white noise without structures, we understand that pixel entropy cannot differentiate noise from geometric shapes with max values as the reviewer suggested. However, since white noise will lead to consistently large pixel entropy in all sliding windows, geometrical structures would exhibit much variable distribution of pixel entropy across windows. Thus, we can also differentiate with standard deviation ($\sigma$) of local pixel entropy when performing LPESW as in **table 2**.
>
> **Table 1: Relationship between maximum values of local pixel entropy (both normalized to [0,1]) and local geometrical entropy ($r^{2} = 0.85$) for various shapes (lines, angles, L-, T-, X- junctions, and jagged edges).**
>
> |**Various shapes** |**Max local pixel entropy** | **Max local geometrical entropy** |
> |-------------------|----------------------------|------------------------------------|
> |line 1            |0.56                         | 0.43                               |
> |line 2            |0.56                         | 0.43                               |
> |angle 1           |0.81                         | 0.87                               |
> |angle 2           |0.79                         | 0.86                               |
> |angle 3           |0.77                         | 0.87                               |
> | L-junction       |0.78                         | 0.74                               |
> | T-junction       |0.78                         | 0.64                               |
> | X-junction       |0.78                         | 0.84                               |
> | jagged edges     |1.00                         | 1.00                               |
>
> **(please see the next official comment)**

---

> ### Author Response · Authors · 2024-08-12
> **Response to Official Comment 2**
>
> **Table 2: Standard deviation values of local pixel entropy and directly calculated global GE can both identify noise from various shapes (lines, angles, L-, T-, X- junctions, and jagged edges).**
>
> |**Various shapes** |**$\sigma$ of local pixel entropy** | **Global geometrical entropy (bits)** |
> |------------------|----------------------------|------------------------------------|
> |White Noise       |4.34e-04                     |NaN                                 |
> |line 1            |0.13                         |  0                                 |
> |line 2            |0.13                         | 0                                  |
> |angle 1           |0.17                         | 1                                  |
> |angle 2           |0.16                         | 1                                  |
> |angle 3           |0.17                         | 1                                  |
> | L-junction       |0.13                         | 1                                  |
> | T-junction       |0.14                         | 2.37                               |
> | X-junction       |0.16                         | 2.73                               |
> | jagged edges     |0.21                         | 4.65                               |
>
> We appreciate the reviewer's careful and instructive comments and plan to incorporate GE in revisions and future work. This will enhance the robustness of our analysis and clarify how pinwheel centers can sensitively respond to the complexity of natural images.
>
> **Model's response to binary edge maps**: We used binary edge maps to isolate geometric complexity by eliminating luminance and contrast. This allowed us to focus on detecting complex features inherent in natural scenes. When images are binarized, texture complexity often results in variations in local geometric edges and shapes. These variations can be captured by the measure of local pixel entropy with sliding window, which reflects the local intensity distribution's complexity.
>
> **The natural images response pattern for salt-and-peppers**: Due to the restrictions on uploading figures, we have instead provided the data in a tabular format (**table 3**). The latency for salt-and-pepper neurons on boundaries is significantly lower than that for neurons in other areas (P < 0.0001) (Corresponding figure is available in anonymous GitHub repository).
>
> **Table 3: Latency for salt-and-pepper neurons on boundaries and neurons in other areas.**
>
> | **Salt-and-peppers' neuron type**    | **Mean latency (ms)** | **Standard deviation (ms)** |
> |--------------------------------|---------------------|--------------------------------|
> | Neurons on boundaries          | **2.832**          | 2.998                           |
> | Neurons in other areas         | 5.137              | 4.626                           |
>
> **Filtered images**: The BSDS 500 images we used are whitened, aligning with our model's training process (Olshausen and Field, 1996, *Nature*) (citation 36 in our paper). Whitening reduces nearby pixel correlation by down-weighting low-frequency content, which typically dominates and causes correlations. This process also enhances important image features, such as edges and contours, to preserve structural details. Additionally, whitening attenuates high frequencies, which often correspond to noise, thereby reducing noise from being introduced into the image.
>
> **PCs tuning for particular types of edge structure**: Figure 4e in our paper shows that PCs have broader orientation tuning curves than iso-orientation domains, which may enable them to detect T-junctions and corners, as demonstrated by Ming Li et al. (2019, *Sci.Adv.*) and Erin Koch et al. (2016,*Nat. Commun.*). We analyzed the acute angles formed by the primary and secondary peaks in the tuning curves (**Table 4**), revealing that PCs prefer larger acute angles, closer to orthogonal (90°), indicating a preference for orthogonal junctions. While this result doesn't distinguish between "L" and "T" junctions beyond their angle, we suggest that higher visual cortices like V2 and V4 handle such distinctions, as shown by Tianye Wang et al. (2024, *Nat. Commun.*) and Anna W. Roe et al. (2012, *Neuron*).
>
> **Table 4: Probability distribution of preferred adjusted acute angles in pinwheel centers.** (Available in anonymous GitHub)
>
> | Adjusted acute angle range (degrees) | Probability (%) |
> |-----------------------|-----------------|
> | 0 - 9                 | 0               |
> | 9 - 18                | 0               |
> | 18 - 27               | 1.30            |
> | 27 - 36               | 0               |
> | 36 - 45               | 3.90            |
> | 45 - 54               | 6.49            |
> | 54 - 63               | 5.19            |
> | 63 - 72               | 14.29           |
> | 72 - 81               | 22.08           |
> | 81 - 90               | 46.75           |
>
> **We would like to express our sincere gratitude for your time and effort in reviewing our paper.**

---

### Author Rebuttal · Authors · 2024-08-07

We greatly thank the reviewers for their valuable advice and comments, which are very helpful for us to further improve this work. We are especially encouraged by the recognition from the reviewers:
1. The findings are quite interesting and novel. The modeling approach seems well-designed.
2. This work can be a good candidate for a saliency detector and gives some interesting quantitative predictions.
3. Their model is interesting and matches experiential data across species. The paper is very well-written and their analysis is novel and interesting. The paper solves an interesting problem combined with biophysics and neuroscience.
4. The question is of high interest. The paper is good for a thorough investigation of previous works and good bio-plausibility of the model.

There were some unclear points in the paper that may have caused confusion. To address these unclear points, we have made thorough revisions. The significant changes are summarized as follows:
1. We have validated the efficiency of PCs in response to edges again by testing with real image data, in accordance with the suggestions of Reviewer TFr2.
2. We have investigated more into the relationship between our model and biological ground truth, following the suggestions of Reviewer Boae.
3. We have verified with experimental data and compared it with previous models to validate our model, as suggested by Reviewer FA4k.
4. We have accomplished a DNN implementation to our model SESNN performing object recognition to elucidate the functional significance of differences in response latency between pinwheel centers and iso-orientation domains (IODs), based on the recommendations of Reviewer indp.

---

> ### Author Response · Authors · 2024-08-14
>
> **Related Works**
>
> **Functional Roles of Pinwheel Structures Revealed by the SESNN Model**: Traditional models, such as the self-organizing map (Kohonen, 1982) and computational approaches like on-off models (Miller, 1994; Jang et al., 2020; Song et al., 2021; Najafian et al., 2022) and related ANNs (Margalit et al., 2023; Chizhov and Zaitsev, 2021; Lufkin et al., 2022), lack the dynamic and temporal fidelity needed to realistically simulate the emergence of pinwheel structures in the visual cortex. To address these limitations, we propose the SESNN model, integrating retinotopy data (Srinivasan et al., 2015; Scholl et al., 2013; Tehovnik et al., 2007; Niell and Stryker, 2008), detailed morphological data (Tao et al., 2004; Stepanyants et al., 2009; Amatrudo et al., 2012), and CMF (Veit et al., 2014; Tehovnik et al., 2007; Van Beest et al., 2021) to enhance biological fidelity. The proposed self-evolving spiking neural network (SESNN) model effectively simulates macaque cortical organization and pinwheel development within the orientation preference map (OPM). Additionally, our findings reveal that overlap degree, reflecting similar feedforward inputs from identical RGCs to neighboring neurons, positively correlates with the retino-cortical mapping ratio (Jang et al., 2020), distinguishing different V1 organizational patterns.
>
> **PCs and IODs in Neural Processing Hierarchies**: Our results show that pinwheel centers (PCs) and iso-orientation domains (IODs) exhibit distinct neural activity waves, leading to varied responses to contour complexity in spatial-temporal dynamics. PCs, with broader multi-orientation selectivity, respond first to complex contours, followed by IODs, which process simpler edges. PCs correlate strongly with contour saliency, indicating a more prominent role in visual processing than IODs. In contrast, rodents with salt-and-pepper organizations exhibit less pronounced contour saliency. While PCs are associated with higher-order processing due to delayed responses (Li et al., 2019; Song et al., 2020), this delay is likely stimulus-dependent. Previous studies show IODs experience cross-orientation suppression under complex stimuli (Koch et al., 2016), whereas PCs, with their broader tuning, are less affected.
>
> **PCs as Geometric Saliency Detectors**: The SESNN model reveals that PCs have broader orientation tuning and less selectivity for complex contours compared to IODs, which exhibit sharper tuning and cross-orientation suppression, favoring simpler edges (Koch et al., 2016; Li et al., 2019; Ferster and Miller, 1996; Sato et al., 2016; Nauhaus et al., 2008; Blakemore and Tobin, 1972; Bonds, 1989). PCs' excitation reduces cross-orientation suppression, making them more responsive to contour complexity, especially with binary input. This contrasts with rodents having salt-and-pepper organizations, where distinct contour complexity saliency is lacking. Prior studies (Koch et al., 2016; Li et al., 2019; Song et al., 2020) indicate that PCs have delayed response latency, suggesting higher-order processing. This is attributed to the use of drifting grating stimuli that preferentially activate IODs. However, these studies omit temporal neural data within pinwheel structures. The SESNN model corroborates physiological findings that IODs and PCs prefer simple and complex orientation stimuli, respectively.

---

### Decision · Program_Chairs · 2024-09-25

**Decision:**

Accept (poster)

**Comment:**

The paper proposes a novel self-evolving spiking neural network (SESNN) that develops either salt-and-pepper or pinwheel organization reminiscent of primary visual cortex. The author propose that pinwheel centers respond more effectively to complex patterns than salt-and-pepper populations. The reviewers recognize the work as novel and innovative, especially the fact that the model replicates the pinwheel structure of monkeys and that it illustrates a functional difference between pinwheel and salt-and-pepper organization. However, the reviewers also criticized the entropy-based measure used for evaluation and this criticism could not be fully resolved during the author-reviewer discussion. Overall, the positives outweigh the negatives and I suggest the paper for acceptance. However, I ask the authors to tune down their claims that rest on the contested metric and add a discussion of its limitations to the final version of the paper.